# A pH-dependent cluster of charges in a conserved cryptic pocket on flaviviral envelopes

Lorena Zuzic[1,2], Jan K Marzinek[1], Ganesh S Anand[3,4], Jim Warwicker[5], Peter J Bond[1,3]*

[1]Bioinformatics Institute (A*STAR), Singapore, Singapore; [2]Department of Chemistry, Manchester Institute of Biotechnology, The University of Manchester, Manchester, United Kingdom; [3]Department of Biological Sciences, 16 Science Drive 4, National University of Singapore, Singapore, Singapore; [4]Department of Chemistry, The Pennsylvania State University, University Park, United States; [5]School of Biological Sciences, Faculty of Biology, Medicine and Health, Manchester Institute of Biotechnology, The University of Manchester, Manchester, United Kingdom

*For correspondence:
peterjb@bii.a-star.edu.sg

Competing interest: The authors declare that no competing interests exist.

**Abstract** Flaviviruses are enveloped viruses which include human pathogens that are predominantly transmitted by mosquitoes and ticks. Some, such as dengue virus, exhibit the phenomenon of antibody-dependent enhancement (ADE) of disease, making vaccine-based routes of fighting infections problematic. The pH-dependent conformational change of the envelope (E) protein required for fusion between the viral and endosomal membranes is an attractive point of inhibition by antivirals as it has the potential to diminish the effects of ADE. We examined six flaviviruses by employing large-scale molecular dynamics (MD) simulations of raft systems that represent a substantial portion of the flaviviral envelope. We utilised a benzene-mapping approach that led to a discovery of shared hotspots and conserved cryptic sites. A cryptic pocket previously shown to bind a detergent molecule exhibited strain-specific characteristics. An alternative conserved cryptic site at the E protein domain interfaces showed a consistent dynamic behaviour across flaviviruses and contained a conserved cluster of ionisable residues. Constant-pH simulations revealed cluster and domain-interface disruption under low pH conditions. Based on this, we propose a cluster-dependent mechanism that addresses inconsistencies in the histidine-switch hypothesis and highlights the role of cluster protonation in orchestrating the domain dissociation pivotal for the formation of the fusogenic trimer.

## Editor's evaluation

Using state-of-the-art molecular dynamics simulations, the authors discuss the potential binding sites of drug molecules to the flaviviral envelope. Moreover, using constant pH simulations, they discuss the functional relevance of a cluster of ionizable residues in a cryptic site at the domain interface. These results have provided novel mechanistic insights into the pH-dependent conformational changes of the envelope protein and cryptic binding sites in the envelope protein that can be targeted for inhibiting viral infection.

## Introduction

Flaviviruses belong to a family of enveloped positive-sense, single-stranded RNA viruses which are transmitted by arthropod vectors, predominantly mosquitoes and ticks. The family includes human

pathogens such as dengue (DENV), yellow fever (YFV), Zika, tick-borne encephalitis (TBEV), Japanese encephalitis (JEV) and West Nile virus (WNV), which can be subdivided into a neurotropic and non-neurotropic group of infectious agents (*Gaunt et al., 2001*). The non-neurotropic group of flaviviruses, associated with haemorrhagic disease, is primarily transmitted via *Aedes* mosquitoes and is mainly present in the tropical and subtropical regions of the world (*Bhatt et al., 2013*; *Kraemer et al., 2015*). In recent years, partly as a result of global warming, the mosquito distribution has been rapidly expanding into temperate regions and is now co-distributed with the majority of the human population (*Brady et al., 2012*; *Kamal et al., 2018*; *Kraemer et al., 2019*).

Flaviviral particles consist of an internal nucleocapsid and a surface-facing envelope, which is composed of a membrane and two types of embedded viral proteins. An envelope (E) protein is exposed to the surface of the virus and appears as a first point of contact between the virus and host immune system, whereas a smaller membrane (M) protein, which plays a role in viral maturation, localizes underneath the E protein and is concealed from the exterior. Both proteins are embedded in the viral membrane via transmembrane (TM) helices, and together they form stable heterotetrameric ($E_2M_2$) complexes (*Zhang et al., 2013a*; *Kostyuchenko et al., 2013*; *Sirohi et al., 2016*; *Sevvana et al., 2018*). In the mature virion, 90 $E_2M_2$ units are organised in a tight-knit herringbone-like pattern with three heterotetramers arranged in parallel to form a raft. In total, 30 rafts form a complete viral envelope and are likely a functional unit for immune recognition (*Zhang et al., 2013a*; *Sevvana et al., 2018*; *Figure 1*).

During the viral entry process, the virion is internalised via clathrin-mediated endocytosis (*Mosso et al., 2008*; *Acosta et al., 2008*) and is contained within endosomes. A gradually acidifying environment inside the compartment (~pH 5–6.5; *Randolph and Stollar, 1990*) triggers a large-scale conformational change within the envelope whereupon E proteins dissociate from their $E_2$ dimeric state, which lies flat on the surface of the virus, and instead form trimeric spikes with conserved fusion loops positioned at the tips (*Modis et al., 2004*). The fusion loops anchor the virus to the host membrane, effectively creating a bridge that spans the gap between the viral and endosomal membranes. A series of putative conformational changes in the spike trimer catalyse membrane fusion, penetrating the endosome and allowing for the viral nucleocapsid with its RNA genome to be released into the host cytoplasm (*Harrison, 2015*).

The identity of the exact residues responsible for triggering this dramatic sequence of conformational changes remains unclear, but the majority of research has been focussed on several conserved histidine residues as the most likely pH-sensing candidates, guided by the principles of the 'histidine switch hypothesis'. This hypothesis presupposes histidines to be pH-sensing residues responsible for triggering conformational changes in viral fusion proteins that are dependent on pH. The hypothesis hinges on the property of the histidine side chain having its $pK_a$ close to physiological pH ($pK_a = 6.0$), and therefore presumes that only histidines are able to change protonation state under the relatively mild pH changes occurring inside the living cell. Site-directed mutagenesis experiments on recombinant subviral particles of TBEV demonstrated that a single His317 mutation (residue numbers corresponding to DENV2) was sufficient for inhibiting a membrane fusion step, likely because of impaired dissociation of E protein dimers. Double His244/His282 mutants had a reduced stability of fusogenic trimers, which also impaired fusion (*Fritz et al., 2008*; *Stiasny et al., 2011*). A mutation in His144 could not yield stable subviral particles and its effect on the fusion step could therefore not be determined. However, subsequent research on WNV reporter virus particles has brought into question the histidine-switch hypothesis of the pH-dependent conformational change (*Nelson et al., 2009*). Even though mutations of His144 and His244 significantly affected the fusion process, substitutions to non-titratable Met/Asn and aromatic residues, respectively, resulted in partial rescue of infectivity.

One of the best researched flaviviruses is DENV (serotypes 1–4) which poses a serious health risk because it can lead to dengue haemorrhagic fever or dengue shock syndrome (*Gould, 1998*). More serious forms of the disease have been linked to antibody-dependent enhancement (ADE), where a first infection produces anti-DENV antibodies specific against one strain, which then imperfectly cross-react with another strain during a subsequent infection (*Halstead, 1979*; *Balsitis et al., 2010*; *Guzman et al., 2013*). This unexpected detrimental effect of the host's immune system has been linked to subtle differences in E protein sequences among DENV serotypes, inefficient viral maturation resulting in a high number of immature particles presenting in the extracellular space, as well as to a striking heterogeneity of DENV particle morphology observed under various environmental

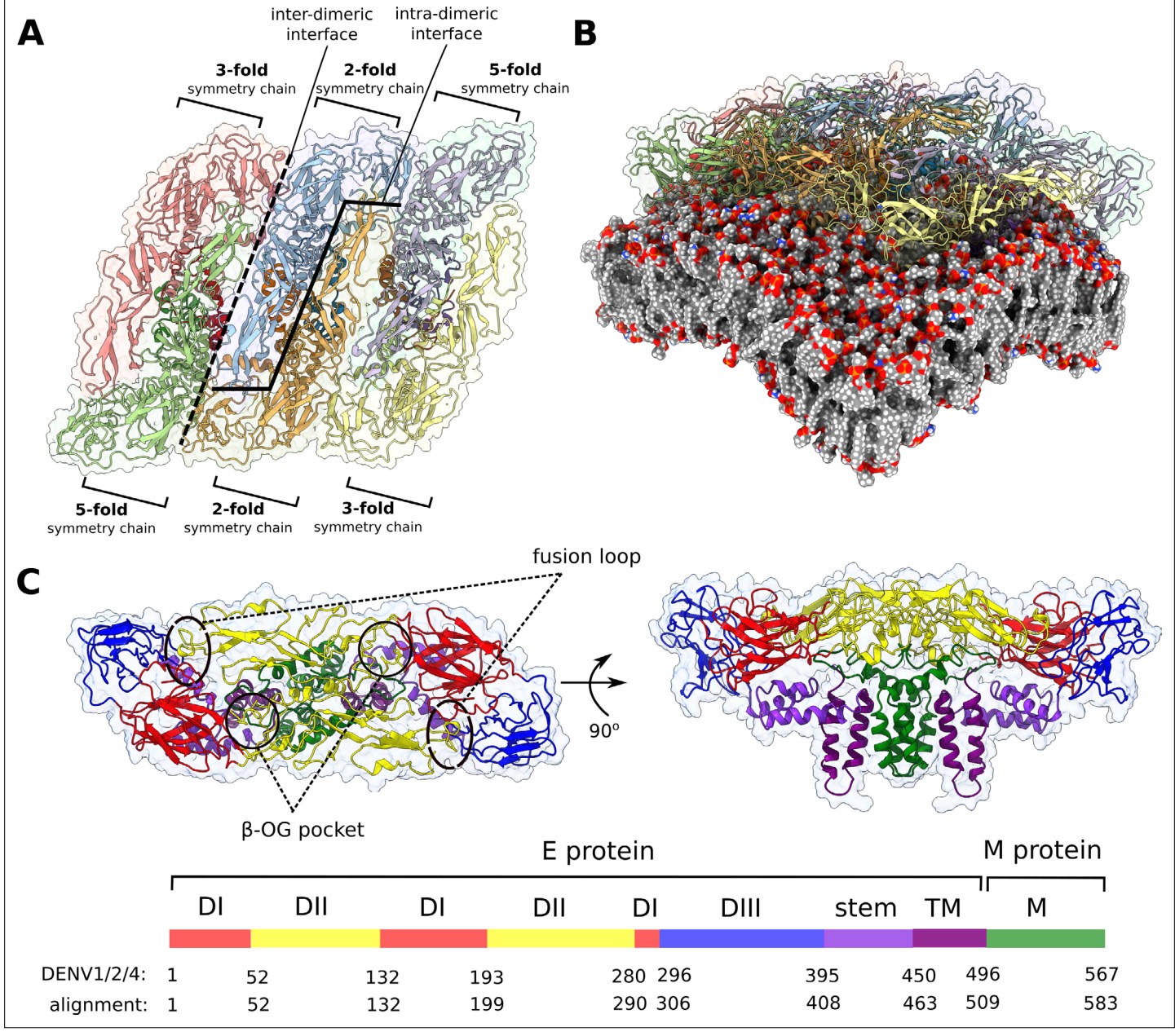

**Figure 1.** Flaviviral raft structure and domain organisation. (**A**) A top-down view on the flaviviral envelope protein raft consisting of three $E_2M_2$ heterotetramers organised in a parallel fashion. Chains differ in the context of their immediate environment and are described in terms of the symmetry fold, which is repeated across the entire viral particle. In this model, the majority of interchain contacts is retained for the central dimer (consisting of two 2-fold symmetry chains), while the edge 3-fold symmetry chains are not bordering neighbouring rafts as in the full virion. (**B**) The curvature of the envelope membrane is imposed by that of the protein rafts. See also *Figure 1—figure supplement 1*. (**C**) Top and side views of the $E_2M_2$ heterotetramer shown in ribbon representation and coloured by functional domains. Protein surface is shown as a transparent outline. The six flaviviral strains differ in length, and residue numbers on the key correspond to the alignment of all strains and to DENV1, 2, and 4. By convention, individual residues mentioned in the text are always matched with DENV2 numbering.

The online version of this article includes the following figure supplement(s) for figure 1:

**Figure supplement 1.** An atomistic model of a DENV2 raft and its curved membrane overlaid with a cryo-EM map of a DENV2 envelope (PDB: 3J27; EMDB: EMD-5520).

conditions associated with temperature and pH (*Modis et al., 2004*; *Junjhon et al., 2010*; *Fibriansah et al., 2013*; *Zhang et al., 2013b*; *Lim et al., 2019*; *Morrone et al., 2020*; *Fibriansah et al., 2021*). The mechanism of ADE is most likely multifaceted (*Morrone and Lok, 2019*; *Wirawan et al., 2019*), but predominantly includes anti-DENV antibodies interacting with the Fcγ receptor (FcγR) leading to enhanced viral internalisation into the endosome, from which the virus can escape by utilising its fusogenic spikes (*Balsitis et al., 2010*; *Rey et al., 2018*; *Williams et al., 2013*). The aggravation of disease caused by antibodies puts into question conventional routes of fighting viral infection, which usually involve vaccines designed to promote humoral immunity and a strong antibody response to the presence of a virus. Sanofi's *Dengvaxia*, the only dengue vaccine currently in use, has been shown to be serotype-dependent and can lead to ADE in those who have not contracted the virus prior to vaccination (*Guy et al., 2017*).

An alternative to antibody-reliant approaches for fighting viral infections are antiviral drugs which may act upon various stages of the viral life cycle. There have been numerous attempts to apply drug design to target flaviviruses, and they have usually focussed on the non-structural viral proteins involved in viral replication or polyprotein processing, as well as some host targets involved in viral assembly (reviewed in *Boldescu et al., 2017*). Recent studies have demonstrated that cyanohydrazone molecules bind to a pocket on the E protein to inhibit viral entry at micromolar concentrations for DENV2, Zika, and JEV (*de Wispelaere et al., 2018*; *Li et al., 2019*). The pocket in question is located at the domain I-domain II interface of the E protein, is cryptic in nature, and was fortuitously observed in crystal structures of the DENV2 soluble portion of the E protein (sE) bound to a single *n*-Octyl-*β*-D-Glucoside (β-OG) detergent molecule (*Modis et al., 2003*). The effectiveness of these antivirals in blocking the early stages of the viral life cycle without causing ADE encourages exploration of other possible druggable sites on flaviviral envelopes. It also highlights the potential of utilizing drug design approaches targeting cryptic sites that may be of functional importance to the virus.

One of the methods focussed on detection of potential drug-binding sites is a co-solvent simulation approach that exploits the property of preferential binding of small organic molecules onto protein surfaces with which they can establish favourable interactions. Such surfaces are known as hotspots and their detection is a critical step for rational drug design (*DeLano, 2002*). To that aim, the probes resembling common functional drug moieties (such as benzene or hydrocarbons) are preferably used, as their binding location and mode of interaction may be instructive for a subsequent drug design step (*Guvench and MacKerell, 2009*; *Tan et al., 2016*). On top of being a frequent component of drug-like molecules (*Kolb and Caflisch, 2006*), benzene is also apolar in nature. This property is useful for uncovering cryptic sites which are at least partially hydrophobic in character and therefore only transiently appear in a detectable 'open' conformation (*Cheng et al., 2007*; *Schmidtke and Barril, 2010*; *Oleinikovas et al., 2016*; *Kuzmanic et al., 2020*).

Here, we use large-scale co-solute molecular dynamics (MD) simulations to explore the behaviour of six flaviviral rafts (DENV1-4, YFV, and Zika) and reveal conserved cryptic pockets in the presence of benzene probes. To that aim, we establish a raft model as a realistic representation of the viral envelope that successfully reproduces the features of curvature and interchain contacts, while being attainable in an all-atom level description and thus providing a precise insight into physicochemical properties applicable to the envelope as a whole. The dynamics of the ~400,000 atom raft systems are explored using a total simulation sampling time of over 14 μs. We investigate a landscape of potentially druggable sites on the envelope surface, compare pocket conservation across viral strains in relation to their sequence similarity, and propose conserved and functionally important sites with potential for drug targeting and inhibition of the early stages of the viral life cycle. Finally, we elaborate on the functional importance of a proposed cryptic site by highlighting a conserved charged cluster of residues that includes the key histidine residue, His144, explore its behaviour in different pH-environments using a constant pH-simulation approach, and propose a cluster-dependent mechanism of DI-DIII dissociation essential for formation of the fusogenic trimer.

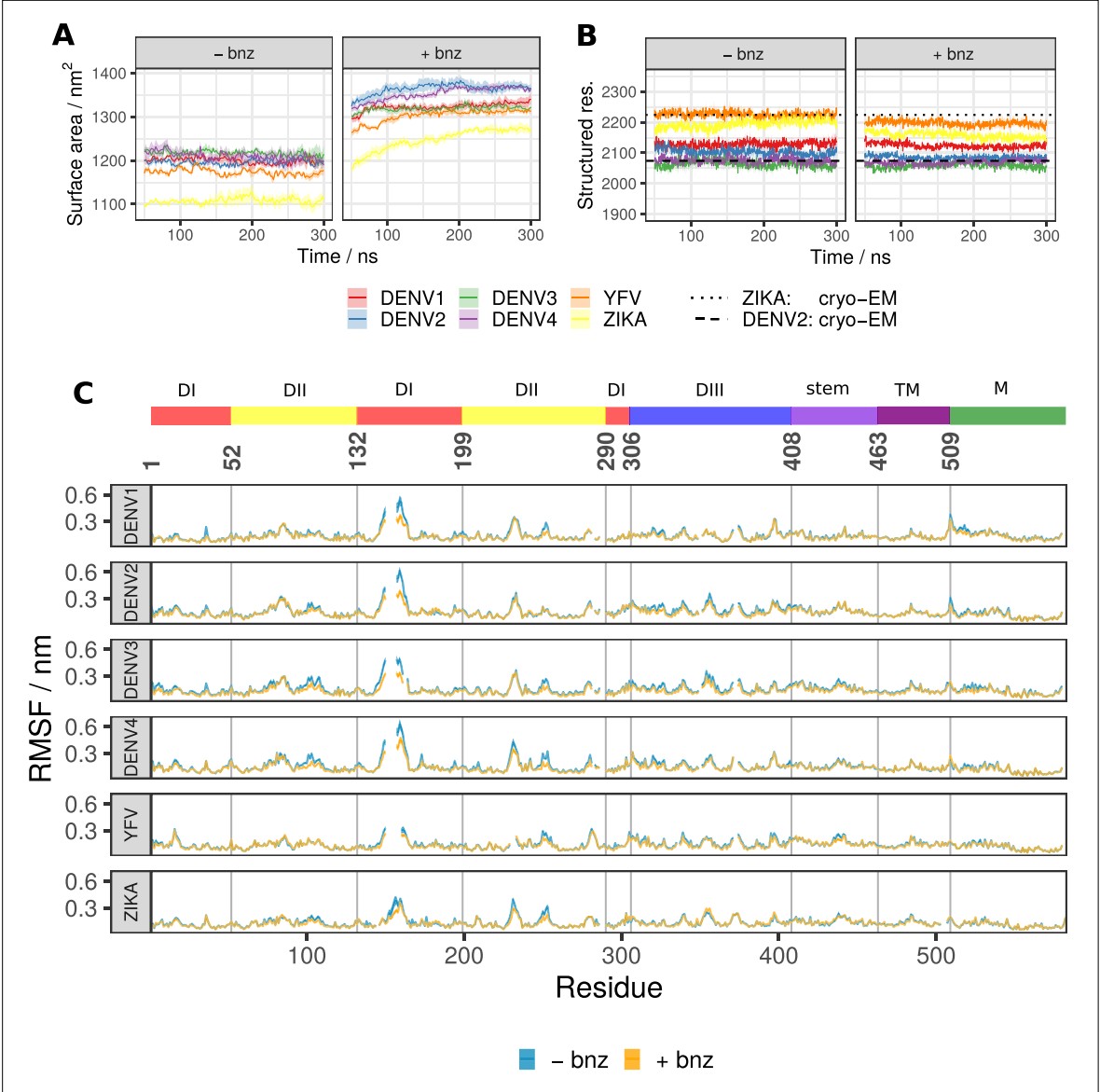

**Figure 2.** Global raft properties for all six viral serotypes showing the effect of benzene probes added to the solvent. (**A**) The total SASA across all viral systems increases with the addition of hydrophobic benzene probes to the solvent. The first 50 ns of each simulation was considered as equilibration and therefore omitted from the analysis. (**B**) The number of residues participating in secondary structure (defined as constituent elements of α-helices, β-sheets, β-bridges, or turns) remains largely unaffected in benzene systems and closely corresponds to the values describing the experimentally derived cryo-EM structures. (**C**) The RMSF values describing whole-residue fluctuations are similar across solvent types and viral serotypes. The gaps in lines are due to the sequence alignment which accounts for differences in length or deletions/insertions between different flavivirus envelope proteins, ensuring that the corresponding residues are vertically aligned. All values displayed in panels are averaged across repeats and, in the case of RMSF, across all chains of the system. Standard error is shown as a transparent ribbon around the mean value.

## Results and discussion
### Benzene binding increases solvent protein accessibility without affecting the secondary structure

High concentrations of apolar probes can potentially lead to undesirable unfolding events by invading and disrupting the hydrophobic core of the protein, resulting in a shift towards biologically irrelevant conformations (*Schmidt et al., 2019*). Thus, in the current work, we first confirmed that the addition of benzene to flaviviral raft systems increases the total SASA without affecting protein secondary structures (*Figure 2A–B*). The unperturbed secondary structure suggests that the increased SASA

of the sampled conformations of the rafts is not due to protein unfolding, but instead is a result of subtle side chain rearrangements, benzene incorporation into protein-protein interfaces, and cryptic pocket opening events. Additionally, very similar RMSF profiles for water-only and benzene simulations demonstrated that the overall dynamic fluctuations were comparable in behaviour despite the differences in solvent composition (*Figure 2C*). Highly dynamic loop regions (such as the N153 glycan loop) had fractionally lower RMSFs compared to water-only simulations. This effect might be due to stabilisation imposed by benzene interacting with the disordered loop regions and the ordered surface beneath, thus providing a stable interaction surface for the flexible portions of the protein.

## Benzene binding patterns are sequence-dependent

Flaviviral serotypes, although similar in structure, exhibit subtle serotype-specific differences in envelope dynamics. Even though the polyprotein sequence similarity between the presently explored flaviviruses is relatively high (42% for all six serotypes and 71% for DENV serotypes only), a large number of mutations localise on the surface-exposed portions of the envelope protein (*Figure 3—figure supplement 1A–B*). This is unsurprising in the context of viral evolution, as mutations arise in the surface-exposed areas in order for the virus to evade the host immune response. However, certain exposed regions of the E protein are highly conserved because they play specific functional roles in the viral life cycle and are therefore less likely to accumulate mutations.

Shared sites of benzene binding reflect residue conservation patterns on the E protein and mostly localise around the conserved fusion loop and at the pr-binding interface (*Figure 3* and *Figure 3—figure supplement 1C*). The fusion loop (residues 98–111) is conserved among flaviviruses as it inserts into the endosomal membrane, which is essential for catalysing the membrane fusion event leading to genome release into the host cell cytoplasm (*Allison et al., 2001*). Similarly, the pr portion of the prM protein protects the fusion loop from premature insertion during the maturation process, and its interaction surface on the E protein is therefore also conserved to a high degree (*Heinz et al., 1994*; *Figure 3—figure supplement 1A and C*). It is interesting to note that protein-protein interfaces have similar characteristics to drug-binding hotspots (*DeLano, 2002*; *Mattos and Ringe, 1996*; *Zerbe et al., 2012*) and the methods used for detecting one are also likely to detect the other type of protein surface. This was well demonstrated by benzene mapping, as we found that benzene molecules preferentially interacted with the pr-binding interface, highlighting it as a potential hotspot.

Overall, though, the surface patterns of benzene binding and, consequently, increased SASA, were variable and serotype-dependent (*Figure 3*). The external E protein surface is mostly unconserved and correspondingly showed a significant degree of variability in benzene binding patterns across serotypes. Notably, a central portion of DII was benzene-depleted (conservatively, residues 58–65, 120–125, 215–235, 252–267), suggesting that it is unlikely to be an attractive drug-binding hotspot. The width of this depleted band was sequence-dependent, with it being the narrowest in DENV4 and the widest in YFV. There are currently no known flaviviral antibodies that specifically target the central band of the raft, corroborating the observation that poor benzene binding sites are also unlikely to be involved in the formation of protein-protein interfaces. The benzene interaction sites slightly differed depending upon the position of the chain within the raft (*Figure 1A*). Protein-protein interfaces that were positioned on the raft edges were more easily accessible to benzene molecules than the interfaces engaged in interchain interactions. Despite this, benzene mapping was highly similar across chains and displayed clear serotype-specific patterns of binding.

The areas of frequent benzene binding also matched increases in SASA, showcasing the role of benzene in exposing protein sites that predominantly remain hidden in water-only simulations. The TM regions of the E proteins and the M proteins showed little increase in SASA, which is explained by the fact that those regions are sequestered within the membrane and are, for the most part, out of reach of benzene molecules.

Correlation analyses of serotype properties, especially SASA, recapitulated phylogenetic relationships between the viruses (*Figure 3—figure supplement 1D*). Serotypes that are more closely related generally exhibited more similar dynamic behaviour, highlighting the importance of sequence conservation in yielding shared characteristics such as SASA, benzene binding, or pocket formation. Interestingly, not all residue-based properties showed an equally high degree of correlation with sequence – notably, the RMSF correlations were poorly reflective of flaviviral evolutionary relationships, suggesting a more complex interplay of properties causing residue fluctuations than sequence

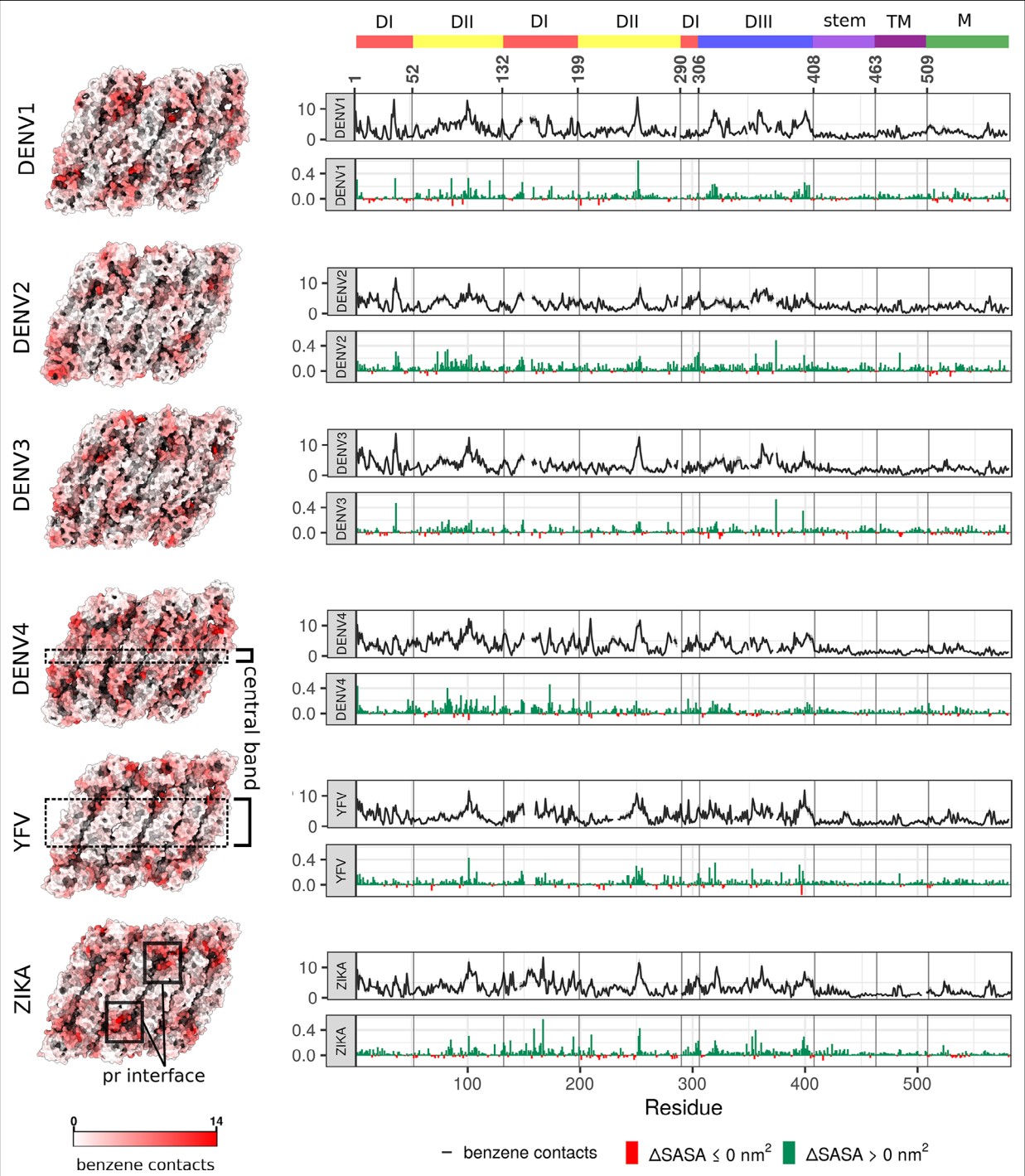

**Figure 3.** Benzene contacts and changes in SASA shown for all six viral serotypes. (Left) The average number of benzene contacts per residue is shown on viral raft structures. The values for individual chains are averaged across repeats. For patterns of residue conservation and phylogenetic relationships between strains, see *Figure 3—figure supplement 1*. (Right) In each upper graph, the mean value of benzene contacts per residue averaged across all chains and repeats is shown in black line, with standard error displayed as a grey ribbon. In each lower graph, the average change in SASA is shown in nm (*Bhatt et al., 2013*), calculated as ($SASA_{+bnz}$ - $SASA_{-bnz}$), with an increase shown in green and decrease in red. As with *Figure 2*, the gaps in lines are due to flavivirus-specific sequence alignment.

The online version of this article includes the following figure supplement(s) for figure 3:

**Figure supplement 1.** Residue conservation and phylogenetic relationships of flaviviruses correlated with simulation properties.

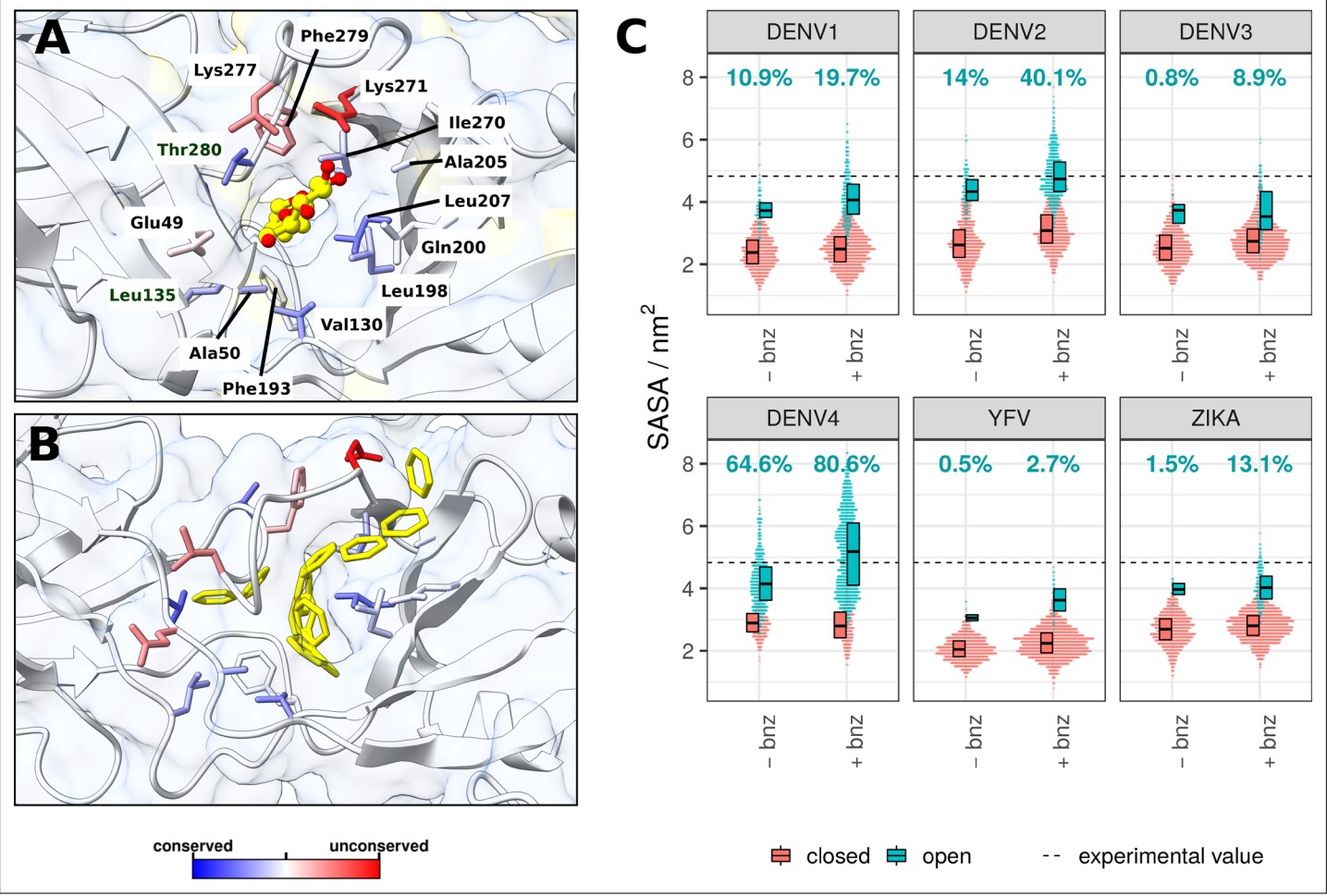

**Figure 4.** β-OG pocket in the presence of the β-OG ligand or benzene, and its SASA across viral serotypes. (**A**) The β-OG detergent molecule co-crystallised inside the cryptic pocket of the DENV2 sE protein (PDB code: 1OKE) (*Modis et al., 2003*) . The protein is shown as a combination of white ribbon and transparent surface representations. The β-OG molecule is shown in yellow sticks, with oxygen atoms highlighted in red. The residues surrounding the ligand are accentuated in stick representation and coloured based on their conservation scores calculated in Consurf (*Ashkenazy et al., 2016*). Residue labels correspond to DENV2. (**B**) A representative simulation snapshot of benzene molecules occupying the β-OG pocket in the DENV2 system. Benzenes are shown in yellow stick representation. Other elements of the system are visualised in the same way as in panel A. (**C**) Cumulative SASA for all residues lining the pocket (labelled in panel A) across six flaviviral serotypes. The data was visualised as a swarm plot, where a SASA measurement for each sampled frame was represented as a horizontally stacked non-overlapping point. The spread of points therefore describes the distribution of SASA measurements for systems in absence (-bnz) or presence of benzene (+bnz). Pocket SASA values are shown separately depending on the solvent composition. Open and closed conformations of the pockets were differentiated by applying a *k*-means clustering algorithm (details in the Methods section). The percentage of the β-OG pocket observed in an open conformation is specified for each simulation system. The indicated experimental SASA value of the pocket is that measured for the crystal structure (panel A) averaged across the two chains. See also *Figure 4—figure supplement 1*.

The online version of this article includes the following figure supplement(s) for figure 4:

**Figure supplement 1.** β-OG pocket properties of SASA and benzene contacts shown for individual viral strains and chain types.

alone (*Figure 3—figure supplement 1D*). Instead, the RMSF values showed remarkable similarity across all six viral serotypes (*Figure 2C*).

## The open/closed state ratio of β-OG pocket is specific to the viral serotype

The β-OG cryptic pocket was first detected in DENV2 sE protein crystal structures (*Modis et al., 2003*) where a single detergent molecule intercalated within the hydrophobic pocket located at the DI-DII interface of each chain (*Figure 4A*). The binding pocket was absent in apo-structures as the position of

the *kl* loop obscured the entry to the site (*Figure 1C*). Functionally, the DI-DII interface is a hinge region and its movement is essential for the E protein to undergo pH-dependent conformational changes leading to the formation of fusogenic trimers (*Zhang et al., 2004*). However, the co-crystallised β-OG molecule was absent from sE dimer structures of other flaviviruses (*Kanai et al., 2006*; *Nybakken et al., 2006*), and even from the E protein structures of the same viral serotype (*Zhang et al., 2004*) despite the addition of the detergent in the crystallisation buffer. Nonetheless, other studies have shown that the β-OG pocket is likely present in at least some flaviviruses. The JEV apo-sE crystal structure showed that the *kl* loop is positioned in a way to reveal a pocket large enough to accommodate β-OG, even in the absence of the ligand (*Luca et al., 2012*). Indeed, pyrimidine compounds and cyanohydrazones (both groups exhibiting inhibitory effects on the fusion events of DENV2, WNV, JEV, and Zika viruses) appear to bind inside the β-OG pocket, as demonstrated by loss-of-binding site-directed mutagenesis and photocrosslinking experiments (*de Wispelaere et al., 2018*; *Li et al., 2019*).

Using the benzene mapping simulation approach on raft systems, we explored the characteristics of the β-OG pocket in different flaviviral serotypes. We observed that the ratios of open/closed states of the pocket differ greatly depending upon the particular virus (*Figure 4C*), with the biggest pocket exposure attributed to DENV4 followed by DENV2, with YFV exhibiting comparatively the lowest pocket SASA even in the presence of benzene. Pocket properties were also affected by solvent composition, with simulations containing benzene manifesting, on average, higher SASA values. Benzene molecules can enter the hydrophobic pocket and transiently interact with residues lining the site (*Figure 4B*). The number of benzenes that occupied the pocket during the course of the trajectory was in direct correlation with the pocket SASA (*Figure 4—figure supplement 1B*), with DENV2 and DENV4 showing the highest propensity for accommodating benzene molecules within the pocket, both in terms of number and frequency. In contrast, YFV β-OG pockets were the least likely to contain any benzene. Therefore, the presence of benzene probes in the system modified pocket exposure and, consequently, its SASA, either by locking the pocket in an open conformation or by inducing the opening of the pocket in the first place.

The observed open/closed ratios of the β-OG pocket were not equivalent for all chains in the system; instead, they were dependent on the position of the chain within the raft. The edge 3-fold chains appeared to have the highest median SASA value, an effect that was in particular exaggerated for DENV2 and DENV4 systems (*Figure 4—figure supplement 1A*). This might be due to the fact that the edge chains were unconstrained by the neighbouring elements on one side, allowing for greater flexibility of the protein and easier opening of the pocket.

We clustered pockets based on SASA and contact properties to separate the sampled conformations into two clusters corresponding to open and closed states (*Figure 4C*). The DENV4 pocket appeared to be predominantly sampled in an open conformation; the DENV2 pocket was in an open state only half as often as compared to DENV4. Contrastingly, DENV1 and DENV3 exhibited very few open conformations. This variability is reflective of a relatively low conservation of residues (53%) among the DENV group of viruses – an attribute that also highlights the potential difficulty in developing pan-flaviviral therapeutics targeting the β-OG site.

The evident differences in pocket exposure can only be partially explained by the differences in the residues lining the pocket, and it is likely that the pocket opening is influenced in the wider context of protein residues outside of the binding site. *de Wispelaere et al., 2018* identified a mutation in DENV2 that is detrimental to antiviral compound binding, even though the point-mutation M196V is outside of the β-OG pocket. These findings suggest that the pocket opening might be allosterically affected by residues outside the binding cavity, either by modifying its shape, frequency of opening, or by affecting the stability of the interactions with its binding ligands.

Remarkably, the SASA value measured for the open β-OG pocket in the crystal DENV2 sE structure (*Modis et al., 2003*) closely corresponded to the mean pocket value in simulated DENV2, suggesting that the benzene simulations successfully sampled biologically relevant open conformations of the pocket that match experiment. Despite some of the serotypes exhibiting predominantly closed states of the pocket even with the addition of benzene, those β-OG pockets are not necessarily absent. Instead, the kinetics of site opening might be serotype-dependent, resulting in pockets of some of the viruses predominantly residing in closed states. Thus, successful inhibition of the fusion events in Zika in the presence of β-OG pocket-binding inhibitors suggests that the pocket is sufficiently druggable, despite its presently observed low fraction (13.1%) of open states.

# The α pocket located on the domain interfaces is conserved and contains a buried cluster of charges

Our previous studies of a single DENV2 $E_2M_2$ heterotetramer identified a novel cryptic pocket (referred to as α) located on domain interfaces (*Zuzic et al., 2020*; *Figure 5*). Here, we found that this pocket was consistently present in all explored flaviviral serotypes and for all raft chains (*Figure 5—figure supplement 1*). The increase in SASA in the presence of benzene was evident for all mapped α pockets, confirming its cryptic character. Interestingly, properties of the α pocket across serotypes appeared to be more consistent than those of the β-OG pocket, and this uniformity could be linked to a higher degree of residue conservation (47% for all explored flaviviruses; 70% for the DENV group). SASA was also less affected by the location of the chain within the raft for the α pocket, likely due to the fact that it is positioned peripherally and is therefore less dependent on intra-raft contacts. We cannot exclude that the dynamics of the α pocket opening might be affected by inter-raft contacts not present in our models.

The pocket is elongated and functionally divided into two parts - the fusion loop interface, representing the interchain interaction surface; and the DI-DIII cryptic pocket segment that is located underneath the N153-glycosylation loop (*Figure 5A*). A closer inspection of the cryptic segment reveals that it is well conserved, with surface residues showing a greater degree of variability, and buried residues appearing overall more conserved. The N153-glycan loop obstructing the entrance to the pocket was highly flexible in our simulations for all DENV serotypes (*Figure 1C*). Although we omitted the glycan component from our models, the simulations were in agreement with the hydrogen-deuterium exchange experiments performed on the whole DENV2 particle that demonstrate a high degree of flexibility in the glycosylation loops (*Lim et al., 2017*). Furthermore, cryo-EM structures of the whole viral particles (*Zhang et al., 2013a*; *Sevvana et al., 2018*; *Renner et al., 2021*), as well as crystal structures of E proteins expressed in insect cells (*Modis et al., 2003*; *Zhang et al., 2004*; *Modis et al., 2005*; *Barba-Spaeth et al., 2016*) indicate that the N153-loop is unstructured. Interestingly, E protein dimers expressed in bacterial systems (*Luca et al., 2012*; *Lu et al., 2019*) appear to have the N153-loop structured as an α-helix, suggesting a role of the glycan moiety in loop secondary structure organisation. The flexibility of the N153-loop indicates that it might aid cryptic pocket opening and transient solvent accessibility of residues lining the α pocket.

The DI-DIII cryptic segment of the pocket contains a conserved His144 residue (*Figure 5A*) that has been implicated in fusogenic pH-dependence (*Nelson et al., 2009*) and viral particle stability (*Fritz et al., 2008*). In a static structure (*Zhang et al., 2013a*), the residue appears to be solvent-inaccessible and with a predicted $pK_a$ outside the biological range ($pK_a < 0$), which would make it an unlikely candidate for a pH-sensing residue (*Figure 6—figure supplement 2A*). However, site-directed mutagenesis experiments indicate its importance in the pH-dependent conformational change (*Nelson et al., 2009*), although its exact role is as of yet unknown. The dynamic exploration of the flaviviral envelopes in our simulations shows that the residue could be transiently exposed to the solvent, concurrent with the cryptic pocket opening events, which in turn might affect its $pK_a$.

Generally, the only residues that have tended to be considered in the context of pH-sensing roles in the flaviviral E proteins have been histidines as their intrinsic sidechain $pK_a$ values (~6.0, as determined from titration experiments of compounds outside the protein environment) can rapidly protonate under relatively mild acidic conditions. However, this approach disregards the effect of the surroundings on residue $pK_a$ (*García-Moreno et al., 1997*; *Isom et al., 2011*) and the fact that the residues, especially buried ones, are located in a low dielectric environment. Under such conditions, $pK_a$ values can be dramatically altered – specifically, Asp/Glu residues have the potential to be upshifted (*Srivastava et al., 2007*) and act as pH-sensing or pH-coupled residues (*Sazanavets and Warwicker, 2015*). When E and M protein residues were explored in the context of their burial, conservation, and potential for being titrated, only six of them fulfilled the criteria and, remarkably, all but one were located on the DI-DIII interface associated with the α pocket region (*Figure 5B*). Four conserved residues – Arg9, Asp42, His144, and Glu368 – were found to closely interact and formed a distinct cluster of charges bridging the DI-DIII interface. In addition, the N-terminus (N-ter) – characterised by its burial and ionisability, and unaffected in terms of conservation – was in close proximity to the residues in question and was also a part of the cluster. The presence of closely coupled residues within the cluster suggests that any pH-dependent protonation event on His144 would have to be considered in the context of other charges

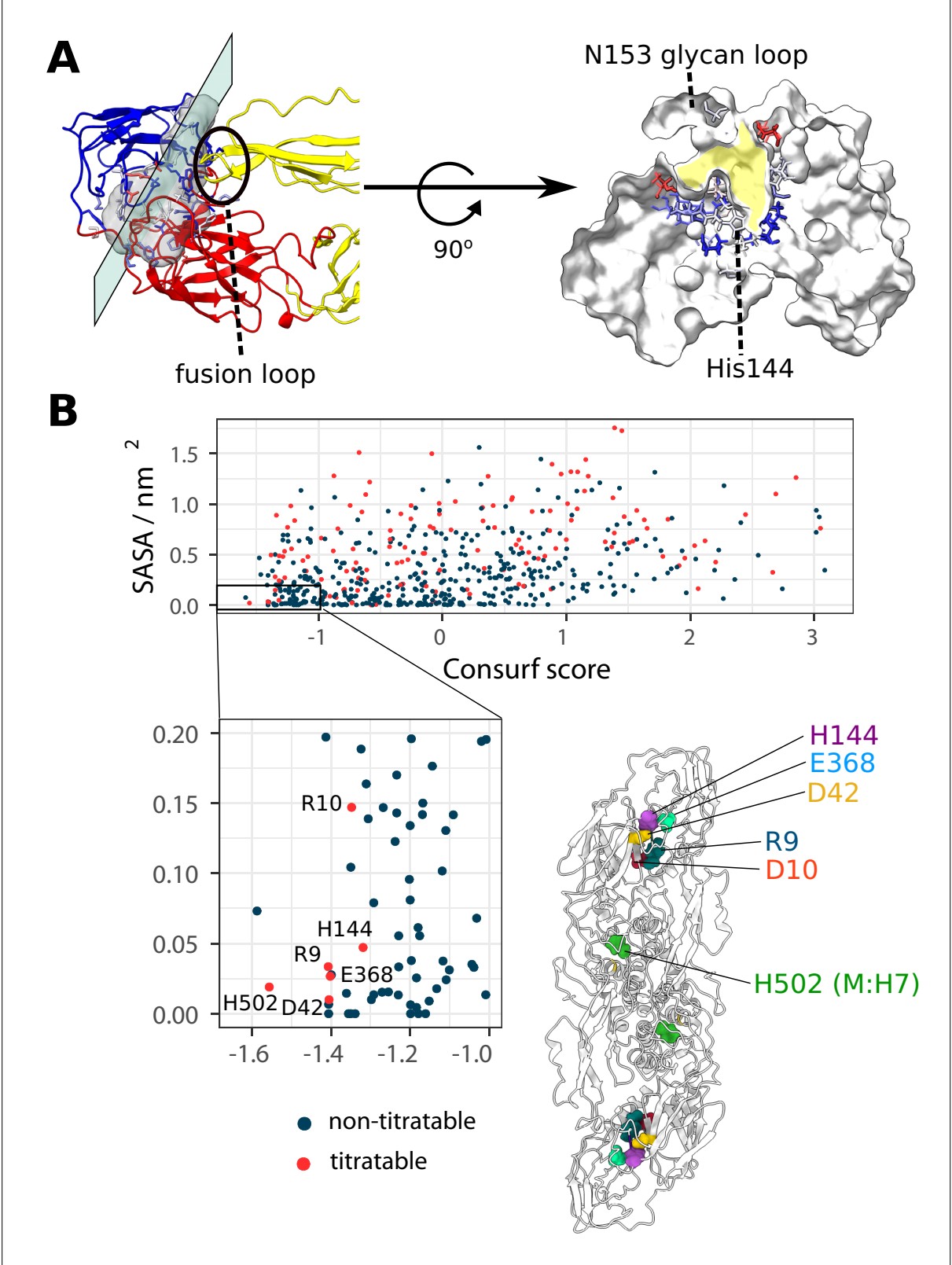

**Figure 5.** The location of the α pocket and the selection of all conserved, buried and titratable residues on the flaviviral envelope, showcasing a charge cluster associated with the pocket. (**A**) Left side: The α pocket density displayed as a transparent grey surface located on the intrachain DI-DIII and interchain DII-DIII interface. The E protein is shown in ribbon representation and coloured according to domain assignments (as seen in *Figure 1*). The plane represents a directional slice of the pocket shown on the right side of the panel. **Right side:** The protein is shown in white surface representation

*Figure 5 continued on next page*

*Figure 5 continued*

and the residues lining the pocket are displayed as sticks and coloured according to their Consurf conservation score. The pocket is highlighted in yellow. (**B**) The selection of E and M protein residues according to the properties of conservation, burial, and ionizability. Residues characterised by a high degree of conservation (Consurf score ≤–1.0) and solvent inaccessibility (SASA ≤0.2 nm$^2$) are shown in the inset. Additionally, titratable residues are labelled and displayed on the $E_2M_2$ structure, where it becomes apparent that almost all selected residues occupy the DI-DIII interface region associated with the α pocket. For α pocket SASA measurements across flaviviral strains, see *Figure 5—figure supplement 1*.

The online version of this article includes the following figure supplement(s) for figure 5:

**Figure supplement 1.** α pocket SASA values across different viral strains and solvent compositions.

within the cluster. The effects of pH on the cluster protonation and dynamic behaviour were next explored in constant-pH simulations.

## Disruption of the α pocket charge cluster at low pH

We performed 0.88 µs of constant-pH simulations on the sE dimer of DENV2 across a broad pH range of 0–9. The convergence of the examined p$K_a$ values is shown in *Figure 6—figure supplement 1*. The focus of our analysis was the α pocket charged cluster (N-ter, Arg9, Asp42, His144, Glu368) and its behaviour at different pH conditions (*Figure 6*). Interestingly, even though preliminary p$K_a$ calculations based on the static structure obtained from cryo-EM predicted His144 to be inaccessible for protonation (*Figure 6—figure supplement 2A*), conformational sampling during constant-pH simulations revealed transient solvent accessibility of the charge cluster that likely allows for histidine protonation at endosomal pH (p$K_a$(His144)=4.84; *Figure 6—figure supplement 2B*).

The disruption of cluster contacts (assessed by measuring the radius of gyration of the cluster residues, with higher values corresponding to overall greater disorder) was assessed in relation to the patterns of charges. Considering that the charge cluster is broken apart in the postfusion trimer (*Modis et al., 2004*), it is expected that a decrease in pH would be accompanied by disruption of the cluster. The cluster underwent a charge shift as consequence of lowering pH (*Figure 6B*), which indeed led to a gradual increase in disorder (*Figure 6A*), most noticeable following a total cluster charge shift from 0 to +3.

In high and neutral pH environments, the cluster was mainly in the $D^-H^0E^-$ state, corresponding to protonated N-ter and Arg9 and deprotonated Asp42, His144 and Glu368, resulting in a total cluster charge of 0. The residues within the cluster were predominantly in close contact and formed a tightly interacting unit on the DI-DIII interface. When assessed across all pH values, this close-contact state dominated the conformational landscape as it occurred in 95% of all sampled structures (*Figure 6C*). Notably, the remaining conformations, characterised by looser residue association, were enriched at positive cluster charges occurring at low pH (*Figure 6—figure supplement 3*). This uneven distribution of conformations across charge states suggests a dissociation effect linked with acidic pH conditions, despite its relatively low occurrence as compared to the compact conformation. The constant-pH simulations did not include titrations of the N-ter or Arg9 and instead contained a fixed +1 charge at all pH values. However, given that the intrinsic p$K_a$ values for both residues are in the basic range (p$K_a$ = 9.6 and p$K_a$ = 12.0) and their predicted p$K_a$ values are 7.4±0.2 and 9.45±1.9, respectively (*Figure 6—figure supplement 2A*), it is likely that both residues remain largely protonated in the acidic environment of the endosome.

Specific patterns of charge distribution across residues had different effects on cluster stability, even when the total charge remained unchanged: $D^-H^+E^-$ (protonated His144) had little effect on cluster disruption, while $D^-H^0E^0$ (protonated Glu368) introduced a charge imbalance effect leading to the disturbance of the overall cluster structure. The cluster at +1 charge was a significant presence in the pH 5–6 range, relevant in the context of the endosomal pH-dependent conformational change (*Figure 6B*).

Transition of the total cluster charge to the +2 value resulted in substantial destabilisation of the DI-DIII interface, specifically in a $D^-H^+E^0$ state (protonated His144 and Glu368). The switch to this charge state was accompanied by a protein-wide conformational change in the sE dimer with DIII shifting upwards from its initial position (*Figure 6C*). The disruption was linked to Arg9 decoupling from the rest of the residues to point away from the cluster. At very low pH (0–2), the cluster exhibited a total charge of +3 ($D^0H^+E^0$, corresponding to all-residue protonation), and this was associated with even greater disruption of cluster contacts and conformational changes that manifested as DIII

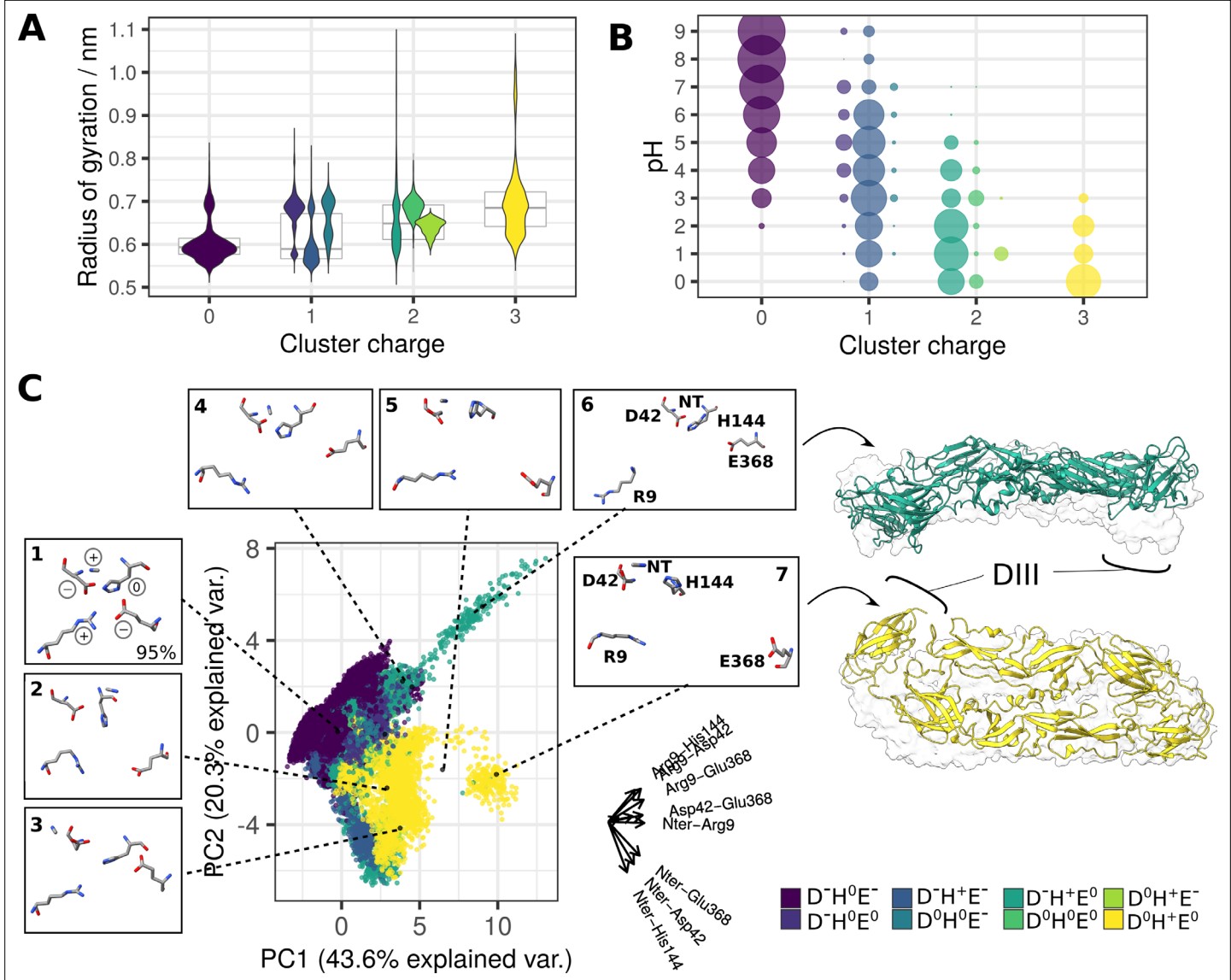

**Figure 6.** The response of the charge cluster to varying pH conditions. For convergence of p$K_a$ values in constant-pH simulations, see **Figure 6—figure supplement 1**. (**A**) Disruption of the charge cluster (assessed via its radius of gyration) in relation to the charge pattern (violin plots in the foreground) and overall cluster charge (boxplots in the background). The measurements were obtained over a range of pH values (pH 0–9). The violin plots represent the distribution of y-values (radius of gyration measurements) for each protonation state, where the total area of violin(s) for each protonation state is constant. The codes in the legend describe residue charges for the residues that were titrated in the simulation system: Asp42, His144, and Glu368. N-ter and Arg9 had a fixed positive charge and were therefore omitted. For predicted p$K_a$ values of selected residues, see **Figure 6—figure supplement 2**. (**B**) Occurrence of distinct protonation states at different pH conditions. Point sizes correspond to the population sizes at each value. D$^-$H$^0$E$^-$ is a dominant pattern of charge in high and neutral pH conditions, whereas the remaining seven patterns (representing +1,+2, and+3 total cluster charge) appear at low pH conditions. See also **Figure 6—figure supplement 3**. (**C**) PCA describing cluster conformations based on the intra-cluster distances. The axes corresponding to the original variables are shown on the right side of the plot (labels for the smallest axes His144-Glu368 and Asp42-His144 omitted for clarity). Clustering analysis of the five residues (details in the Methods section) generated nine distinct conformations which are projected onto the PCA and shown in insets around the plot. Conformations 8 and 9 had a negligible contribution to the total number of structures and were therefore excluded from the visualisation. The snapshots corresponding to structures undergoing a greater conformational change at low pH are visualised in ribbon representation (trajectories shown in **Figure 6—videos 1; 2**). Transparent surfaces represent the starting sE dimer conformation, demonstrating the extent of the conformational shift. Ribbon colours correspond to the specific charge patterns under which the conformational change had occurred (green and yellow for D$^-$H$^+$E$^0$ and D$^0$H$^+$E$^0$, respectively). For SASA of the α pocket across different charge states of the cluster, see **Figure 6—figure supplement 4**. For the effects of benzene on the conserved cluster behaviour, see **Figure 6—figure supplement 5**.

The online version of this article includes the following video and figure supplement(s) for figure 6:

**Figure supplement 1.** Convergence of p$K_a$ values.

*Figure 6 continued on next page*

*Figure 6 continued*

**Figure supplement 2.** Prediction of p$K_a$ values for selected E protein residues using static structures and MD simulations.

**Figure supplement 3.** Occurrence of conformations at different charge states.

**Figure supplement 4.** α pocket SASA values across different protonation states (violins) and charge states (boxplots) of the conserved cluster.

**Figure supplement 5.** Benzene contacts and its effects on the charge cluster.

**Figure 6—video 1.** The top-down view of the DIII dissociation in DENV2 sE dimer under low pH conditions.

https://elifesciences.org/articles/82447/figures#fig6video1

**Figure 6—video 2.** The side view of the DIII tilt in DENV2 sE dimer under low pH conditions.

https://elifesciences.org/articles/82447/figures#fig6video2

shifting away from DI (*Figure 6C*). Notably, this change appears to be different from the conformational switch observed in the D$^-$H$^+$E$^0$ state as it was predominantly linked to Glu368 distancing from the remainder of the cluster residues.

Even though we observed protein-wide conformational changes only in more extreme pH environments than are observed within the endosome, it is evident that at least one associated pattern of charge (D$^-$H$^+$E$^0$) can occur at pH 5, suggesting that this route of DI-DIII disruption at endosomal pH is possible. Alternatively, the observed increase in intra-cluster distances at total charge zero suggests that it might also lead to DI-DIII dissociation in the context of better sampling or a more expansive representation of the viral particle surface.

Based on our findings, we propose that the drivers of conformational change at low pH are more complex than the simplistic histidine switch hypothesis, which limits possibilities for pH-sensing residues exclusively to histidines. As shown in our simulations, the solitary His144 protonation event has little effect upon overall sE dimer stability; instead, the conformational change appears to be inextricably linked to an interconnected network of conserved titratable residues that jointly contribute to the pH-dependent formation of the fusogenic spike. The cluster effect might also explain incongruous results in loss-of-function experiments which have shown that E proteins with an H144N mutation can still undergo the fusion event (*Nelson et al., 2009*), suggesting that His144 plays a part, but is not indispensable for triggering trimer formation.

## Limitations

In this work, we modelled the portion of the flaviviral envelope with the aim of detecting cryptic pockets on its protein components, but we omitted 1–2 glycan molecules (at position N67 for DENV1-4 only, and at N153 for all addressed flaviviral species). At the time of conducting the research, the benzene mapping method had not been verified against systems containing sugar moieties, and we hypothesised that the shielding effect of glycans might hinder effective benzene exploration of the protein surface. Furthermore, the experimental data available regarding glycan types on each glycosylation site is scarce and is dependent on the viral species and the examined host (*Hacker et al., 2009*). However, the α pocket that contains the cluster of charges implicated in pH-dependent conformational change is located underneath the N153-glycan loop and as such, it is expected to at least partially affect the dynamics of pocket opening. This aspect of flaviviral envelope dynamics remains unaddressed at this time, and a topic for the future.

The membrane was modelled under geometric and compositional constraints. The curvature of the membrane was induced by introducing position restraints on Cα atoms of the TM helices. The membrane consisted of three phospholipid species only, and it did not include sphingolipids or cholesterol, both of which are suggested to be functionally important for flaviviral infection (*Martín-Acebes et al., 2014*; *Carro and Damonte, 2013*). Taking into consideration the simplicity of the membrane composition model, as well as the application of position restraints in the lipid-proximal areas of proteins, detailed analysis of protein-lipid interactions was not pursued. Instead, the membrane served as a low-dielectric and solvent-excluding 'shield' that mimicked the approximate environment of the viral envelope and disallowed benzene to interact with the underside of the modelled raft.

The constant-pH simulations were performed on a truncated model of the E protein dimer, and the titrations were performed only on a small number of ionisable residues (specifically, the identified cluster charge residues and conserved histidines). The constant-pH method utilised the Generalised Born solvation model to attempt protonation, while the conformational sampling was performed in

an explicit water environment. These aspects of the constant-pH simulation setup may be limiting in terms of accuracy of the used model and the comprehensiveness of the protonation sampling. Therefore, the future continuation of this research may involve constant-pH simulation approaches that allow for simulations of bigger systems and with titration coordinates applied to all acid-ionisable residues, such as a recently developed method reported by *Aho et al., 2022*.

## Perspectives on cryptic hotspot discovery in flaviviruses

Prophylactic or therapeutic approaches for fighting flaviviral disease that utilise vaccines or antibodies might be suboptimal because of the danger of ADE. Instead, antiviral drugs that could target the flaviviral life cycle could be an effective alternative. Specifically, a potential drug targeting the early stages of the life cycle would not have to be membrane-soluble as it would target the viral particle in the extracellular environment. Furthermore, inhibiting the fusion step would contribute to prevention of ADE, because it would pre-empt viral escape from the endosome prior to degradation. Hence, antiviral drugs would not only offer protection on their own, but could also act in concert with vaccines. The inhibitory action of antiviral drugs by binding to the β-OG pocket has already been demonstrated for a range of flaviviruses (*de Wispelaere et al., 2018*; *Li et al., 2019*), likely by preventing a hinge-like movement of the DI-DII interface necessary for formation of the fusogenic trimer. Similarly, targeting drug-like molecules to the α pocket could modulate charge cluster behaviour, effectively disrupting the pH-dependent dissociation of DI-DIII that is an essential step in the endosomal conformational change. Consistent with the cluster disruption and domain dissociation observed under low pH conditions, the surface area of the α pocket also increases with the net charge of the cluster (*Figure 6—figure supplement 4*). A drug molecule binding inside the pocket and stabilising it – both under neutral or low pH conditions – could conceivably prevent domain dissociation and pH-induced conformational changes of the viral envelope.

His144 is the most accessible to benzene probes in all examined flaviviral serotypes and thus likely to be a primary interaction moiety for potential antiviral drugs (*Figure 6—figure supplement 5A*). Our simulations show that benzene interactions with His144 also lead to an increased disorder within the cluster (*Figure 6—figure supplement 5*), which is an effect that would need to be considered in the context of drug design. Nonetheless, the proximity of other cluster residues suggests that all of them have the potential to play a role in drug-binding and in turn be modulated in their response to a low pH environment.

## Conclusions

Our simulation studies of DENV1-4, YFV and Zika rafts provide a detailed perspective of the dynamic behaviour of flaviviral envelopes in the presence of benzene. We elaborated on a wider context of shared hotspot characteristics and potentially druggable cryptic pockets that could be utilised in the inhibition of the early stages of the viral life cycle. We have demonstrated that conserved surfaces share not only residue identity, but also similar patterns of benzene binding and increased SASA, highlighting the importance of residue conservation in the context of detection of shared binding hotspots. We also provided a detailed view of the β-OG pocket behaviour in different flaviviral species that exhibited a high degree of serotype specificity when examined for pocket opening events.

Furthermore, the α pocket was mapped across flaviviral serotypes and revealed a relatively consistent profile in terms of conservation, size and opening kinetics. It contained a cluster of conserved ionisable residues, and its position at the domain interfaces prompted further exploration of its role in pH-dependent conformational changes using constant-pH simulations. The conserved cluster displayed increased disorder at lower pH values and interdomain dissociation (a prerequisite for the formation of the fusogenic trimer). Protonation-driven disruption of the DI-DIII interface suggests its critical role in the endosomal pH-dependent conformational change. We also hypothesise that the cluster as a whole – as opposed to the usually considered histidine residues only – plays a role in the conformational shift, corroborating previous experimental observations that histidines alone might not be the only pH-sensing residues responsible for the orchestration of the pH-dependent conformational switch. Collectively, our findings provide a link between the sequence-dependent dynamic behaviour of hotspots and the mechanistic overview of their functional roles, and offer a comprehensive foundation for rational drug design targeting the envelope of flaviviruses.

## Methods

### Benzene mapping of flaviviral rafts

#### Raft modelling

A raft encompasses a large surface area of the viral envelope and therefore exhibits a distinct curvature in its quaternary structure, which manifests itself in angled contacts between neighbouring E proteins (*Figure 1B*). In comparison, crystal structures of ectodomain dimers (*Modis et al., 2003*; *Lu et al., 2019*; *Dai et al., 2016*) are invariably assembled outside the confines of a viral envelope, resulting in a 'flat' conformation of the dimeric structure. Such crystal structures are thus unable to fully replicate native interchain contacts present in the spherical particle, highlighting the necessity of using cryo-electron microscopy (cryo-EM) structures to produce a reliable raft assembly. Currently, there are only two published cryo-EM structures of the mature DENV particle at atomic resolution, both of which are of the DENV2 serotype (*Zhang et al., 2013a*; *Renner et al., 2021*). We selected a raft segment from a viral envelope cryo-EM structure solved at 3.6 Å resolution (PDB code: 3J27) and used it as a model for the DENV2 raft. We then created the remaining DENV serotypes (DENV1, 3, and 4) by mutating individual residues in Modeller 9.19 (*Sali and Blundell, 1993*), based on the DENV2 raft template. Sequence differences between YFV and DENV were deemed too great to create a reliable model of YFV based on the DENV2 template alone (*Figure 3—figure supplement 1B*). Instead, we used an existing crystal structure of the YFV ectodomain (PDB code: 6IW4; *Lu et al., 2019*) and modelled the missing loops (residues 269–272) with Modeller 9.19. Next, we used homology modelling based on the DENV2 template to create the missing TM regions and M proteins. Finally, the chains were aligned with the cryo-EM-derived DENV raft to reproduce the quaternary arrangement and curvature of a spherical viral particle. The Zika raft structure was taken from the existing cryo-EM Zika virus structure (PDB code: 6CO8) solved at 3.1 Å resolution[10].

Protein rafts were embedded within a membrane (see below) and equilibrated during a series of simulations with position restraints of gradually decreasing force constants applied to the stem and TM regions of the proteins. Position restraints on the TM regions were necessary to retain the curvature of the raft. All structures were initially equilibrated for 200 ns with position restraints on the backbone atoms of stem and TM regions (E protein residues: 396–495; M protein residues: 21–72; numbering scheme corresponds to DENV2) using a force constant of 1000 kJ mol$^{-1}$ nm$^{-2}$, followed by another 200 ns equilibration with weaker position restraints ($F_c$ = 50 kJ mol$^{-1}$ nm$^{-2}$) on the TM region Cα atoms only (E protein residues: 450–495; M protein residues: 39–72; numbering scheme corresponds to DENV2).

#### Membrane modelling

We used CHARMM-GUI (*Sunhwan et al., 2008*; *Wu et al., 2015*) to embed the DENV2 raft in a membrane containing 1-palmitoyl-2-oleoyl-phosphatidylcholine (POPC), 1-palmitoyl-2-oleoyl-phosphatidylethanolamine (POPE), and 1-palmitoyl-2-oleoyl-phosphatidylserine (POPS) lipids in 6:3:1 ratio, which reflected a simplified membrane composition of flaviviral envelopes expressed in a C6/36 mosquito cell line (*Perera et al., 2012*; *Zhang et al., 2012*). The produced membrane was initially modelled as a flat sheet, which resulted in lateral raft dimers being incorrectly submerged in the bilayer. In order to induce membrane curvature, we simulated the raft-membrane system with position restraints applied on all backbone atoms of the raft. Within ~400 ns of the simulation, the membrane gradually curved and accommodated the shape of the raft (*Figure 1—figure supplement 1*), with the TM regions transversing the bilayer and the amphipathic stem helices localising between hydrophilic headgroup and hydrophobic tail regions of the membrane (*Figure 1B*). In total, the membrane was equilibrated for 785 ns of unrestrained simulation time.

#### Benzene mapping simulation setup

The strategy using benzene probes for mapping pockets in membrane-bound proteins was adapted from our previous work (*Zuzic et al., 2020*). A massless virtual site was placed at the centre of mass of a benzene molecule, which was used as a point of repulsion between hydrophobic benzene molecules so as to prevent aggregation. Additionally, the same site was used as a point of repulsion between benzene probes and membrane lipids in order to hinder benzene sequestration into the bilayer. Benzene probes were added to a simulation box at 0.6 M concentration (1148 benzene molecules per

system). We tested this concentration for signs of protein unfolding or probe aggregation and found no evidence of either process occurring in our simulation systems (*Figure 2B*). The raft-membrane system containing benzene was solvated in TIP3P water (*Jorgensen et al., 1983*) and neutralised with NaCl, to a final salt concentration of 0.15 M.

### Benzene mapping simulation protocol

All simulations were performed with the CHARMM36m force field (*Huang and MacKerell, 2013*) and in Gromacs 2018.2 simulation software (*Abraham et al., 2015*). Initially, systems were minimized using the steepest descent algorithm until forces converged or reached a maximum value of 100 kJ mol$^{-1}$ nm$^{-1}$. Temperature was set to 310 K and regulated using a velocity-rescaling thermostat (*Bussi et al., 2007*) with a time constant of 0.1 ps and applied on two coupled groups, one containing proteins and lipids, and the other, solute components. The Parrinello-Rahman barostat (*Parrinello and Rahman, 1981*) with semi-isotropic coupling and with a time constant of 2 ps was used to maintain a pressure of 1 atm. The compressibility was set to $4.5 \times 10^{-5}$ bar$^{-1}$ for both $x/y$ and $z$ directions. The leapfrog algorithm integration step was set to 2 fs, while the LINCS constraint algorithm (*Hess et al., 1997*) was applied to all bonds associated with hydrogen atoms in the system. Computation of long-range electrostatic interactions was performed using the particle mesh Ewald (PME) method (*Darden et al., 1993*; *Essmann et al., 1995*), with a 1.2 nm cutoff for real-space interactions and with the integration of a Coulomb potential-shift modifier. The cutoff value for Van der Waals interactions was set to 1.2 nm, with a force-switch modifier applied at 1.0 nm.

The system was run in the canonical (*NVT*) ensemble for 1 ns and with strong position restraints ($F_c = 1000$ kJ mol$^{-1}$ nm$^{-2}$) applied in all three dimensions to heavy atoms of both the protein and membrane components of the system. A subsequent 5-ns *NPT* equilibration involved position restraints on protein heavy atoms only, while the membrane was allowed to relax unrestrained. Finally, a production run was performed for 300 ns, with weak position restraints ($F_c = 50$ kJ mol$^{-1}$ nm$^{-2}$) applied to the TM region Cα atoms of the proteins (E protein residues: 450–495; M protein residues: 39–72; in DENV2 numbering system) in order to retain the curvature in the membrane model. A summary of all simulations and the corresponding number of repeats is listed in *Table 1*. In total, 14.4 µs of production runs were generated.

## Constant-pH MD simulations on DENV2 soluble E protein dimer

For constant-pH MD simulations we used a soluble portion of the E protein (sE) in a dimeric form (residues 1–395). The structure of the sE was taken from the DENV2 cryo-EM structure (PDB: 3J27)

**Table 1.** Flaviviral envelope raft simulations in presence or absence of benzene in the simulation system.

| Strain | Solvent[*] | Time / ns | Repeats |
|---|---|---|---|
| DENV1-PVP159 | water | 300 | 3 |
| DENV1-PVP159 | 0.6 M benzene | 300 | 5 |
| DENV2-NGC | water | 300 | 3 |
| DENV2-NGC | 0.6 M benzene | 300 | 5 |
| DENV3-H87 | water | 300 | 3 |
| DENV3-H87 | 0.6 M benzene | 300 | 5 |
| DENV4-H241 | water | 300 | 3 |
| DENV4-H241 | 0.6 M benzene | 300 | 5 |
| YFV-17D | water | 300 | 3 |
| YFV-17D | 0.6 M benzene | 300 | 5 |
| ZIKA-H/PF/2013 | water | 300 | 3 |
| ZIKA-H/PF/2013 | 0.6 M benzene | 300 | 5 |

[*]Both types of solvent contained charge-neutralising 0.15 M NaCl.

(*Zhang et al., 2013a*). As the E protein was truncated for the purposes of simulation, we capped the C-terminus with an N-*methyl* amide (NME) group. Residues His27, Asp42, His144, His244, His317, and Glu368 in both chains were selected to titrate. His282, although present in the structure and implicated in pH-dependent conformational change, was excluded from the selection as it is natively interacting with the stem region (not present in this setup) and was therefore in a biologically irrelevant environment. Native disulfide bonds were included in the structure. The sE dimer was placed in an octahedron box and solvated with ~84,000 TIP3P water molecules (*Jorgensen et al., 1983*) and neutralising 0.15 M NaCl.

The simulations were performed with the Amber ff99SB force field (*Hornak et al., 2006*) modified to include additional parameters relevant for constant-pH. Bond radii of Asp and Glu residues were reduced as carboxylate oxygens were defined with two hydrogens on each protonation site. All constant-pH simulations were performed using the Amber20 simulation software (*Case et al., 2005*). The system was initially minimised using a combination of steepest descent and conjugate gradient algorithms performed in 5,000 cycles and with protein backbone position restraints defined with a force constant of 4184 kJ mol$^{-1}$nm$^{-2}$ (10 kcal mol$^{-1}$Å$^{-2}$). The system was then gradually heated to 310 K over 400 ps by applying Langevin thermostat with a collision frequency of 5.0 ps$^{-1}$. The *NPT* equilibration step was performed for 4 ns with Langevin dynamics maintaining a constant pressure of 1 atm defined by a collision frequency of 1.0 ps$^{-1}$. The production steps of constant-pH simulations were run under 10 pH conditions across the scale of pH = 0, 1, 2, … 9. Production runs were simulated for 22 ns each in the *NPT* ensemble, with attempted protonation state changes every 100 simulation steps (0.2 ps). Protonation attempts were performed by Monte Carlo sampling (*Mongan et al., 2004*) based on transition free energies derived from generalised Born electrostatics (*Bashford and Case, 2000*) and with the Metropolis criterion used to determine if the protonation change should be accepted or not. All simulations were run with a 2 fs integration step, applying a SHAKE constraint algorithm (*Ryckaert et al., 1977*) to all bonds involving hydrogen atoms. Short-range electrostatic and Van der Waals interaction cutoffs were both set to 0.8 nm. Long-range electrostatics interactions were calculated using the PME method (*Darden et al., 1993*; *Essmann et al., 1995*).

Constant-pH simulations were run in quadruplicates, resulting in a total production time of 880 ns.

## Analysis procedures

### Sequence analysis

Flaviviral polyprotein sequences were accessed from GenBank using following accession codes: DENV1 strain PVP159: AEM92304.1; DENV2 strain NGS-C: P14340.2; DENV3 strain H87: P27915.1; DENV4 strain H241: Q58HT7.1; YFV strain 17D: P03314.1; Zika strain H/PF/2013: A0A024B7W1.1. Sequence similarity was determined using the Sequence Identity and Similarity (SIAS) software. Multiple sequence alignment of the E and M proteins was performed using MUSCLE (*Edgar, 2004*). Alignment visualisation and generation of the neighbour-joining (NJ) tree was performed using Jalview 2.11.1.4 (*Waterhouse et al., 2009*). Conservation assessment and scoring was carried out using the Consurf Server (*Ashkenazy et al., 2016*), where a multiple sequence alignment was generated using the HMMER homolog search algorithm in the UNIREF-90 database with an E-value cutoff of 0.0001.

### MD simulations analysis

After removing the position restraints from benzene probes, the system required additional equilibration time during which benzene established interactions with protein surfaces. We therefore excluded the first 50 ns of all production runs from the analysis. Simulation properties, including solvent-accessible surface area (SASA) (*Eisenhaber et al., 1995*), secondary structure content (*Kabsch and Sander, 1983*), radius of gyration, select residue distances, root-mean-square deviations (RMSDs) and root-mean-square fluctuations (RMSFs), were analysed using Gromacs 2018.2 analysis tools (*Abraham et al., 2015*). Contact distances between benzene and residues of interest were calculated using VMD 1.9.3 (*Humphrey et al., 1996*). In order to avoid counting contacts with the β-OG pocket residues that occur when the probes are outside the binding site, we counted only benzene proximal to at least two oppositely positioned pocket residues. The initial detection of protein pockets was performed with the MDpocket software (*Schmidtke et al., 2011*). Correlation of raft pocket densities across serotypes was calculated using the Fit in Map tool in ChimeraX 1.0 (*Pettersen et al., 2021*).

Protonation states of all titrated residues were extracted using the *chpstats* tool implemented in AmberTools21 (*Case et al., 2005*). p$K_a$ predictions of titratable residues based on a static structure were calculated on the DENV2 E proteins (PDB: 3J27) (*Zhang et al., 2013a*) in an asymmetric unit using the Finite difference Poisson-Boltzmann/Debye-Hückel (FD/DH) method (*Warwicker, 2004*).

## Statistical analysis

All residue-based analyses across serotypes were performed after alignment, ensuring that the residues corresponded to their shared evolutionary origin. All correlation analyses were calculated with a non-parametric Spearman's rank correlation method. β-OG pocket clustering was performed with a *k*-means clustering algorithm (*k*=2) based on the measurements of pocket SASA and the number of contacts established between all the residues lining the pocket. All pairwise significance levels were determined with a Wilcoxon signed-rank test used in comparison of two non-parametric samples. Cluster analysis of the charge cluster residues, applied to all atoms within the group, was based on the *gromos* algorithm (*Daura et al., 1999*) implemented in Gromacs with a specified cutoff value of 0.26 nm. Principal component analysis (PCA) of the charge cluster descriptors – all combinations of intra-cluster distances – was performed on a scaled and centered sample containing all structures across pH values, chains, and constant-pH simulation repeats. Visualisation of the original variable axes used the *ggbiplot* package in R 3.6.3. Additional data processing, analysis and visualisation were carried out in R 3.6.3.

## Acknowledgements

The research was supported by the A*STAR Bioinformatics Institute (BII) core funds, the A*STAR Graduate Academy (A*GA), and the University of Manchester. PJB, JKM, and GSA acknowledge NRF (NRF2017NRF-CRP001-027) for funding. The computational work for this article was performed on resources of the National Supercomputing Centre, Singapore (https://www.nscc.sg), A*STAR Computational Resource Centre, A*STAR Bioinformatics Institute, and the N8 Centre of Excellence in Computationally Intensive Research (N8 CIR) provided and funded by the N8 research partnership and EPSRC (Grant No. EP/T022167/1). We are grateful to Ana Damjanovic for helpful advice regarding constant-pH simulations.

## Additional information

### Funding

| Funder | Grant reference number | Author |
|---|---|---|
| National Research Foundation Singapore | NRF2017NRF-CRP001-027 | Jan K Marzinek Ganesh S Anand Peter J Bond |
| Agency for Science, Technology and Research | | Lorena Zuzic Jan K Marzinek Peter J Bond |
| University of Manchester | | Lorena Zuzic Jim Warwicker |

The funders had no role in study design, data collection and interpretation, or the decision to submit the work for publication.

### Author contributions

Lorena Zuzic, Data curation, Software, Formal analysis, Validation, Investigation, Visualization, Methodology, Writing - original draft, Writing – review and editing; Jan K Marzinek, Conceptualization, Supervision, Validation, Investigation, Visualization, Writing – review and editing; Ganesh S Anand, Conceptualization, Writing – review and editing; Jim Warwicker, Conceptualization, Supervision, Funding acquisition, Writing – review and editing; Peter J Bond, Conceptualization, Resources, Data curation, Supervision, Validation, Visualization, Writing - original draft, Project administration, Writing – review and editing

Author ORCIDs

Lorena Zuzic http://orcid.org/0000-0002-7834-612X
Jan K Marzinek http://orcid.org/0000-0002-5493-8753
Ganesh S Anand http://orcid.org/0000-0001-8995-3067
Peter J Bond http://orcid.org/0000-0003-2900-098X

Decision letter and Author response

Decision letter https://doi.org/10.7554/eLife.82447.sa1
Author response https://doi.org/10.7554/eLife.82447.sa2

## Additional files

### Supplementary files

• MDAR checklist

### Data availability

The full simulation trajectories have not been uploaded to a sharing server due to the size of the datasets (several terabytes of data in total). However, we have uploaded all simulation setup data such that a researcher could re-run our calculations (i.e. coordinates including initial and final frames, parameters, checkpoints, and portable binary run input files, for a total of 88 simulation systems) at Zenodo: https://doi.org/10.5281/zenodo.7037906. More extensive simulation trajectory files will be made available upon request to the corresponding author, without the need to e.g. submit a project proposal. There will be no restrictions on who can access this data. Code/software used to analyse the data are freely available and in the public domain.

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
