## [Editor Report]

Using state-of-the-art molecular dynamics simulations, the authors discuss the potential binding sites of drug molecules to the flaviviral envelope. Moreover, using constant pH simulations, they discuss the functional relevance of a cluster of ionizable residues in a cryptic site at the domain interface. These results have provided novel mechanistic insights into the pH-dependent conformational changes of the envelope protein and cryptic binding sites in the envelope protein that can be targeted for inhibiting viral infection.

---

## [Decision Letter]

**Decision letter after peer review:**

Thank you for submitting your article "A pH-dependent cluster of charges in a conserved cryptic pocket on flaviviral envelopes" for consideration by *eLife*. Your article has been reviewed by 3 peer reviewers, one of whom is a member of our Board of Reviewing Editors, and the evaluation has been overseen José Faraldo-Gómez as the Senior Editor. The following individuals involved in review of your submission have agreed to reveal their identity: Jana Shen (Reviewer #2); Mikael Lund (Reviewer #3).

Essential revisions:

1) A more thorough analysis of the constant pH simulations, especially whether the cryptic binding site(s) identified with fixed-protonation-state simulations is (are) likely to change when pH dependence is included. Analysis of charge capacitance is also recommended.

2) Further clarification of some methodological details, such as the treatment of membrane.

*Reviewer #1 (Recommendations for the authors):*

Overall, the study has been conducted and analyzed rather carefully. I have only a few questions.

1. Glycan has been shown to be important in several recent SARs-Cov-2 spike protein simulations. The authors didn't include the glycan component explicitly. It will be useful to further comment on potential technical challenges that motivated this approximation and its limitations.

2. Many viral envelop proteins (e.g. the HIV) contain a rather high concentration of cholesterol, which appears to play a major role in modulating not only the membrane property but also the structure and assembly of the envelope proteins. The authors didn't include cholesterol in their model. It will be useful to comment on the potential implications.

3. The cryptic binding sites were characterized mainly by the SASA values. Are there other characterizations that might be more relevant for future drug design, such as the volume of the cavity?

*Reviewer #2 (Recommendations for the authors):*

– A major weakness of the paper appears to be the flawed design of the simulation. The benzene mapping simulations were conducted using fixed protonations which are not specified in the paper, and thus they do not address any pH response. Why wouldn't the authors conduct the constant pH simulations first to determine protonation states and then use these protonation states to conduct conventional MD with benzenes?

– Another major weakness is that the conclusions are not very clear at least in my assessment. The authors claimed the α pocket to the cryptic pocket, but this conclusion can not be discerned from the presented figures. In fact, in Figure 2, 3 and others, the y values of the region around residue 144 is discontinuous. Not sure why. The conclusion of the constant pH part is very vague, and I can't understand what it is exactly.

– Overall, there are too many plots that are tangentially relevant. I suggest they can be moved to SI and instead focus the plots on the data that support the conclusions.

– It is unclear what exactly the histidine switch hypothesis is. The discussion needs to be more specific. Related to this, it is unclear how the determined protonation states address the hypothesis.

– The convergence of protonation state sampling should be included in the SI.

*Reviewer #3 (Recommendations for the authors):*– p5, line 32. One or two commas would elevate readability– p6, line 4. I wonder if "mixed-solvent" is the right term here. The benzene concentration is 0.6 M and is, compared to >50 M water, rather a co-solute.

– Membrane curvature: Curvature is imposed by MD constraints as detailed in the Method section. I understand that the membrane is not the focus of the analysis, but how does this curvature co-exist with the PBC of the simulation box? The scheme implies that the membrane + raft is surrounded by replicas.

– Figure 2A: Why do the +bnz plots initially drift? If part of the equilibration, I would have expected them to start at the -bnz levels.

– Figure 4C: It's not obvious how to interpret the "violin" plots: the distributions have no scale; and showing two mirrored halves seems redundant. The same comment applies to Figure 6A.

p13-14. Here I suggest revising the use of "rate(s)" as it took me a while to realise that the discussion is not about kinetics, but statics. I suspect that to many readers, "rate" would imply a dynamic property. Perhaps "ratio", "quotient", or "fraction" could be alternatives.

– Figure 6: This plot is packed with information and is used to support a detailed discussion about the role of the found charge cluster. I think that it works well. At pH is data in Fig6a obtained? I understand that the PCA is based on the indicated residue-residue distances. Another way to describe charge-charge interactions is via an electric multipole expansion of the cluster charges. Perhaps an analysis involving the dipole and quadrupole moment could be revealing. Merely a thought.

– Cluster response to pH changes. Figure 6a/b analyses the net-charge of the cluster which probes the overall protonation state. To judge how a pH change would affect the cluster, the charge capacitance, C=-^2 should be straightforward to extract. Measuring the charge fluctuations, it can be directly linked to a charge response due to a pH change. See e.g. doi:10.1017/S003358351300005X. Along the same lines, I wonder of fluctuations in SASA or Rg would reveal information of how easily the cluster is perturbed. Finally, I would have liked to see how the PCA would change in the presence of benzene.

p22, line 19: I think it would be useful to know exactly how many benzene molecule. Is it sufficient to saturate the protein surfaces? Or is there a deficit which could affect the number of observed contacts.

p22, Section 4.1.1. I would prefer to have slightly more information about the setup: "semi-isotropic" could be more specific as well as details about update intervals of the barostat and thermostat.

p24, line 17: Does this mean that solvent and other solutes are not part of the (de)protonation acceptance criterion? That is, is all explicit solvent replaced by a continuum? If so, could one not use a much cheaper constant pH scheme for conformational sampling?

p26, line 13: I much appreciate that the authors have made the effort to deposit the electronic material on Zenodo. This is a nice and very helpful gesture to the community!

---

## [Author Response]

Essential revisions:1) A more thorough analysis of the constant pH simulations, especially whether the cryptic binding site(s) identified with fixed-protonation-state simulations is (are) likely to change when pH dependence is included. Analysis of charge capacitance is also recommended.

We conducted a more detailed examination of cryptic pocket behaviour under titration regime of constant-pH simulations and included it in section 2.7 (Perspectives on cryptic hotspot discovery in flaviviruses), supported by Figure 6 —figure supplement 4. We also assessed the convergence of protonation state sampling in Figure 6 —figure supplement 1.

Although we examined using charge capacitance to further elaborate on the effect of the charge cluster protonation on protein conformation, we believe that this analysis would not be appropriate for our specific case. Proton-exchange in the used constant-pH simulation method was allowed only for small number of residues, so the changes in protonation states across the pH values do not accurately describe the changes in the overall charge. We provide more details on this topic in response to the reviewer below.

2) Further clarification of some methodological details, such as the treatment of membrane.

We added clarifications to the Methods section: namely, we specified position-restrained residues in the TM domain (4.1.1 Raft modelling); we included the number of benzene molecules used in our simulation systems (4.1.3 Benzene mapping simulation setup); we also added more details about the used barostat (4.1.4 Benzene mapping simulation protocol). Finally, we reasoned about our chosen system setup (regarding glycosylation and membrane composition) in the 2.6 Limitations section. There we also reflected on the possible limitations of the chosen constant-pH simulation method. Further details are provided below in response to the reviewers.

Reviewer #1 (Recommendations for the authors):Overall, the study has been conducted and analyzed rather carefully. I have only a few questions.

We thank the reviewer for their positive comments, and appreciate the opportunity to reply to the questions/suggestions below.

1. Glycan has been shown to be important in several recent SARs-Cov-2 spike protein simulations. The authors didn't include the glycan component explicitly. It will be useful to further comment on potential technical challenges that motivated this approximation and its limitations.

We thank the reviewer for highlighting the role of glycans in the viral envelopes. It is indeed one of the key components of the envelope, and its omission from the model was due to challenges in using the benzene mapping in the presence of glycans. At that point, the applicability of the benzene flooding approach had been verified only against systems containing protein and membrane components, but not with glycans (since then, we have successfully applied this method to the glycosylated SARS-CoV-2 spike protein (Zuzic et al., 2022: https://doi.org/10.1016/j.str.2022.05.006)). We were also concerned that the addition of glycans would hinder benzene access to the tightly-packed protein surface, necessitate an even bigger simulation box and, consequently, require more benzene probes, which would result in an unmanageable amount of exclusions between all probe molecules. Motivated by this important comment, we have now included a “Limitations” section in our paper and elaborated on the reasoning behind using the unglycosylated form of the E protein.

2. Many viral envelop proteins (e.g. the HIV) contain a rather high concentration of cholesterol, which appears to play a major role in modulating not only the membrane property but also the structure and assembly of the envelope proteins. The authors didn't include cholesterol in their model. It will be useful to comment on the potential implications.

The reviewer makes an excellent point on the weaknesses of the modelled membrane, and we acknowledge the potential drawbacks of our chosen membrane composition. The used lipid composition data was based on the lipidomics analysis of DENV2 expressed in C6/36 mosquito cells (Zhang et al., 2012: https://doi.org/10.1074/jbc.M112.384446), but this research failed to address lipid types other than phospholipids. Other studies have highlighted the abundance of sphingolipids in West Nile virus (Martin-Acebes et al., 2014: https://doi.org/10.1128/JVI.02061-14) and the critical role of cholesterol for DENV infection in mammalian cells (Carro and Damonte, 2013: https://doi.org/10.1016/j.virusres.2013.03.005). Additionally, ordered lipid densities (possibly PE) have been observed in the cryo-EM structure of Spondweni, Zika, and DENV2 envelope (DiNunno et al., 2020: https://doi.org/10.1038/s41467-020-18747-4), and the conservation of the pocket across flaviviral species suggests functional relevance of the bound lipid, possibly linked to envelope stabilisation.

In the benzene mapping segment of this research, the membrane predominantly served to provide a hydrophobic medium for the TM helices, shielding the TMs and the underside of the proteins from interacting with the solvent molecules. Considering that the main aim of the benzene mapping was to examine transient cryptic pockets accessible from the viral surface, we put forward that the membrane appropriately served its purpose in this case.

Nevertheless, in light of the key point highlighted by the reviewer, we have also included a paragraph addressing the membrane composition under the 2.6 Limitations section.

3. The cryptic binding sites were characterized mainly by the SASA values. Are there other characterizations that might be more relevant for future drug design, such as the volume of the cavity?

The volume measurements of the cryptic sites can be problematic due to the fact that: (i) they are highly dynamic and transient in nature, (ii) surface edges of both observed pockets are defined by flexible loops, the positions of which can excessively affect the value of the reported volume; (iii) the pocket-mapping algorithms that are applicable to simulation trajectories are for the most part imprecise, as the pocket coordinates and boundaries are insufficiently well defined. Specifically, pocket-mapping algorithms rely on extrinsic properties — namely, 3D coordinates within a simulation box — to define a pocket of interest and to calculate its volume. This makes the alignment of the pocket (essentially, an “empty space”) within and across trajectories a non-trivial task, particularly taking into consideration the highly dynamic properties of cryptic pockets. Additionally, defining the pocket edges towards the surface is arbitrary and for the most part requires human input to filter excessively mapped or “spilled” pocket edges. In practice, this is a noteworthy issue when attempting to make a meaningful comparison between different simulation systems.

Considering the factors above, we refrained from reporting on volume measurements in this paper, as we deemed them to be too imprecise. Instead, we opted for a more comparable measure of the pocket expanse that is SASA, which circumvents some of these issues by using the measurement relating to the “filled” space (surface areas of the residues lining the pocket). Previously, we reported the volume of the ⍺ pocket (2.20 ± 0.51 nm^3^) based on the smaller DENV2 E_2_M_2_ model (Zuzic et al., 2020: https://doi.org/10.1021/acs.jctc.0c00370).

Reviewer #2 (Recommendations for the authors):– A major weakness of the paper appears to be the flawed design of the simulation. The benzene mapping simulations were conducted using fixed protonations which are not specified in the paper, and thus they do not address any pH response. Why wouldn't the authors conduct the constant pH simulations first to determine protonation states and then use these protonation states to conduct conventional MD with benzenes?

In fact, the initial aim of the study was entirely unmotivated from elucidating the pHdependent mechanism of conformational changes in flaviviral envelopes. Instead, our primary goal was to detect cryptic pockets on the flaviviral rafts, which is why we set out to utilise the benzene mapping method on the raft with fixed protonation states from the start. Since the aim was to discover envelope pockets that could potentially be a target prior to cellular entry, the ionization state of amino acids were set according to the neutral pH of the extracellular environment.

Only after the benzene mapping study was conducted, we observed the unusual cluster of conserved residues in the α pocket that merited further investigation. Therefore, the pH-dependence became relevant post factum, which then motivated our constant-pH simulations to further probe the role of the discovered cluster.

– Another major weakness is that the conclusions are not very clear at least in my assessment. The authors claimed the α pocket to the cryptic pocket, but this conclusion can not be discerned from the presented figures. In fact, in Figure 2, 3 and others, the y values of the region around residue 144 is discontinuous. Not sure why. The conclusion of the constant pH part is very vague, and I can't understand what it is exactly.

The cryptic nature of the α pocket is presented in Figure 5 —figure supplement 1, where we compared the pocket SASA across simulations in the absence (-bnz) and presence of benzene (+bnz). This is also indicated in the main text (under section 2.4 The ⍺ pocket located on the domain interfaces is conserved and contains a buried cluster of charges).

The apparent discontinuities in the figures showing properties measured on a sequence residue-basis are due to the sequence alignment – since different flavivirus envelope proteins vary in length and/or contain deletions/insertions – thereby ensuring that the corresponding key residues are vertically aligned. We attempted to make that point clearer in the figure legends.

We thank the reviewer for pointing out the weaknesses in our explanations of the conclusions regarding constant pH simulations. We have now reformulated our conclusions (section 3) in order to make it clearer for the reader.

– Overall, there are too many plots that are tangentially relevant. I suggest they can be moved to SI and instead focus the plots on the data that support the conclusions.

Although we understand that the title mainly highlights exploration of the pH-dependence of the cluster of charges, the conclusions drawn from this research are multi-faceted, and require understanding of both the benzene-based cryptic pocket discovery, as well as the pH-dependent conformational effect of the charged cluster. The figures follow the “trajectory” of the conducted research and assume that the reader is not entirely familiar with the benzene mapping approach or its implications, which is why three out of six figures are dedicated to addressing this research segment.

We are suggesting that the figures are relevant for understanding the overall results of the research, including model features (Figure 1), benzene mapping method verification (Figure 2), effects of benzene on the raft surface (Figure 3), analysis of the positive control β-OG pocket (Figure 4), main features of the ⍺ pocket (Figure 5), and finally, the effect of pH on the charged cluster behaviour (Figure 6).

– It is unclear what exactly the histidine switch hypothesis is. The discussion needs to be more specific. Related to this, it is unclear how the determined protonation states address the hypothesis.

We thank the reviewer for highlighting this important omission of the histidine-switch hypothesis explanation. We have now included more details of the hypothesis in the introduction, and also reinstated it in the conclusion of section 2.5 (Disruption of the ⍺ pocket charge cluster at low pH) to hopefully make things clearer.

– The convergence of protonation state sampling should be included in the SI.

We are grateful to the reviewer for this important suggestion and we have now included the convergence of pK_a_ values for all examined residues in the Supplementary Information (Figure 6 —figure supplement 1).

Reviewer #3 (Recommendations for the authors):– p5, line 32. One or two commas would elevate readability

Indeed, the sentence was clumsily formulated. We have now split the statement into two parts.

– p6, line 4. I wonder if "mixed-solvent" is the right term here. The benzene concentration is 0.6 M and is, compared to >50 M water, rather a co-solute.

We thank the reviewer for the insightful comment as we had not considered the accuracy of the used term. Indeed, the hydrophobicity of benzene, as well as the ratio of the two components, would most accurately classify benzene as a co-solute. We have introduced this correction in p6, line 12. When addressing a wider group of pocket detection methods that are based on different solvent components (including, but not limited to benzene), we used the term “co-solvent” as a more frequently used and broader description of the approach.

– Membrane curvature: Curvature is imposed by MD constraints as detailed in the Method section. I understand that the membrane is not the focus of the analysis, but how does this curvature co-exist with the PBC of the simulation box? The scheme implies that the membrane + raft is surrounded by replicas.

This is the correct understanding of the simulation setup. The membrane exists in the PBC context, which means that it forms an undulating landscape. This is not ideal or physiologically relevant for the context of viral envelopes (although under certain dengue viral morphologies something like it has been observed), but the membrane analysis was not focused on any great detail, as it was not deemed sufficiently accurate. Instead, the curvature was induced by retaining position restraints on C⍺ atoms of the TM helices in order to preserve accurate inter-chain contacts and angles between envelope monomers, which are the key factors in sampling the external cryptic surface.

– Figure 2A: Why do the +bnz plots initially drift? If part of the equilibration, I would have expected them to start at the -bnz levels.

The first 50 ns that were excluded from the analysis were a part of the production run, during which the benzene probes were free to move and bind onto the protein surface. Therefore, SASA of the +bnz systems drifts towards higher values as compared to the -bnz systems. We opted to exclude this from our plot visualisation in Figure 2A and B to remain consistent with all the other conducted analyses (all of them excluded the first 50 ns of the production run).

– Figure 4C: It's not obvious how to interpret the "violin" plots: the distributions have no scale; and showing two mirrored halves seems redundant. The same comment applies to Figure 6A.

Perhaps the most convenient way to interpret violin plots is to imagine them as vertical density plots in which the area represents the abundance of data points at any given value of y. Unlike “standard” density plots, where the density is represented by an area that occupies only the positive values of the y-axis, violin plots utilise both positive and negative values of the central axis, giving them a symmetrical shape.

Figure 4C is a swarm plot, which means that individual measurements across frames are represented as non-overlapping points which, as a result, approximate a violin plot shape (but does not fully correspond to it). The area covered by data points is therefore affected by the number of sampled points. In 4C, +bnz systems are sampled more as compared to -bnz (they consist of 5 and 3 repeats, respectively), which also means that the total area covered by points in +bnz is bigger as compared to -bnz systems.

This is unlike “true” violin plots (such as Figure 6A), in which the number of samples is not directly correlated with the total area of the violin(s) at a given discrete value of x. In other words, even though the majority of the samples in Figure 6A belongs to a cluster of charge 0, the area of this violin plot is the same as the much less sampled state with cluster charge of +3.

In conclusion, we thank the reviewer for pointing out the issues with interpretability of these plots. We have therefore added clarifications in figure texts (both for Figure 4C and 6A).

p13-14. Here I suggest revising the use of "rate(s)" as it took me a while to realise that the discussion is not about kinetics, but statics. I suspect that to many readers, "rate" would imply a dynamic property. Perhaps "ratio", "quotient", or "fraction" could be alternatives.

We are grateful to the reviewer for this comment, as we believe that the suggested term modification improved the accuracy of our claims. Instead of using the term “rate”, we instead now refer to the “ratio of open/closed states” of the pocket.

– Figure 6: This plot is packed with information and is used to support a detailed discussion about the role of the found charge cluster. I think that it works well. At pH is data in Fig6a obtained? I understand that the PCA is based on the indicated residue-residue distances. Another way to describe charge-charge interactions is via an electric multipole expansion of the cluster charges. Perhaps an analysis involving the dipole and quadrupole moment could be revealing. Merely a thought.

The data in Figure 6A contains cumulative radius of gyration measurements across all simulated pH values (pH 0-9). We have now added this clarification in the Figure text.

We considered expanding our analyses by determining the dipole moment for the cluster, but we were hindered by some analysis implementation issues. Namely, it is unclear to us how to extract information about partial charge changes of our titratable residues from the constant-pH simulations (implemented in Amber). Calculating the dipole moments from simulations with fixed charges would be relatively easy, as the dipole moment would be (only) a function of atom positions. However, the constant-pH simulations allow for the changes of partial charges of atoms included in the titratable residue. As far as we understand, to calculate this we would require information about the fluctuations in partial charge for all relevant atoms and across all trajectory frames. Obtaining this information from the run simulations was, sadly, not obvious to us.

– Cluster response to pH changes. Figure 6a/b analyses the net-charge of the cluster which probes the overall protonation state. To judge how a pH change would affect the cluster, the charge capacitance, C=-^2 should be straightforward to extract. Measuring the charge fluctuations, it can be directly linked to a charge response due to a pH change. See e.g. doi:10.1017/S003358351300005X. Along the same lines, I wonder of fluctuations in SASA or Rg would reveal information of how easily the cluster is perturbed. Finally, I would have liked to see how the PCA would change in the presence of benzene.

We thank the reviewer for pointing out an interesting approach to quantifying molecular charge fluctuations expressed as a protein property of capacitance, which we have not considered before. We used the Eq. 12 from the linked article to calculate protein charge capacitance (Author response image 1), as our constant-pH simulations were run in a titration regime (pH 0-9).

**Author response image 1. sa2fig1:** sE protein dimer capacitance as a function of pH. The charge fluctuation was allowed on twelve residues of the simulation system (six on each monomer): His27, Asp42, His144, His244, His317, Glu368.

However, we are reluctant to include this analysis in the publication, as the main limitation of the constant-pH method is the fact that the proton-exchange was allowed only for a limited number of titratable residues (as detailed in the Methods section 4.2. Constant-pH MD simulations on DENV2 soluble E protein dimer). This results in an incorrect picture of the net charge on the simulated protein, as only a selection of residues underwent titration, whereas the other residues were assigned a fixed protonation state. Furthermore, the envelope model used for constant-pH simulations was truncated (it contained only the soluble E protein dimer), which would also be a limitation in assessing the capacitance of a much larger multimer forming the viral envelope.

The constant-pH simulations are therefore useful for elaborating on the shorterrange effects of cluster residue (de)protonations and their effect on inter-domain stability. However, they cannot give us an accurate picture of the overall variance in mean charge, as it is a property that hinges upon the “titration regime” of the whole protein – or in this case, the viral envelope.

The mapping of cluster changes in the presence of benzene cannot be provided within the scope of this study, as benzene-flooding simulations were performed under the fixed protonation state regime only. We instead included an additional analysis of the ⍺ pocket, namely the mapping of the pocket surface area across different protonation states of the cluster (Figure 6 —figure supplement 4). As the constant-pH simulations were run with a reduced-size envelope model and only on DENV2, the absolute numbers of pocket SASA are not as informative as with the raft simulations (Figure 5 —figure supplement 1). Nonetheless, the general trend of pocket expansion is consistent with our observations of cluster disruption and domain dissociation under low-pH conditions. Using this information, we have further elaborated on the implications of the ⍺ pocket targeting by drug molecules and their potential mechanism of inhibition in Section 2.7 Perspectives on cryptic hotspot discovery in flaviviruses.

p22, line 19: I think it would be useful to know exactly how many benzene molecule. Is it sufficient to saturate the protein surfaces? Or is there a deficit which could affect the number of observed contacts.

We added the information about the number of added benzene molecules (1148 per system) in the manuscript under the section 4.1.3 Benzene mapping simulation setup.

Our best guess is that the saturation level was not reached with this number of benzenes (in preliminary studies of DENV2 rafts with 0.8 M benzene, we did not observe unfolding effects), but there was a concern that the addition of too many benzene molecules would have undesirable effects in terms of membrane behaviour, subsequent unfolding, or the abundance of nonspecific “noise” contacts between proteins and probes. Therefore, we did not pursue the upper bounds of the saturation value (which is unknown for our specific system).

p22, Section 4.1.1. I would prefer to have slightly more information about the setup: "semi-isotropic" could be more specific as well as details about update intervals of the barostat and thermostat.

We have added more detail about compressibility and time constants of both thermostat and barostat to the manuscript (under the section 4.1.4 Benzene mapping simulation protocol).

p24, line 17: Does this mean that solvent and other solutes are not part of the (de)protonation acceptance criterion? That is, is all explicit solvent replaced by a continuum? If so, could one not use a much cheaper constant pH scheme for conformational sampling?

This understanding of the (de)protonation criterion being assessed in the generalised Born implicit solvent model is correct, and we acknowledge it as a downside of the applied method. We did not consider use of the cheaper implicit solvent for conformational sampling (as implemented by Baptista et al., https://doi.org/10.1063/1.1497164), as the main bottleneck of the constant-pH simulation approach was not the computational expense, but method availability. The computational cost of the simulation was not considered as a major factor in our method selection, and we decided to use the explicit water model where possible. This was presumed to provide enhanced credibility in conformational sampling. Therefore, the method which combines an explicit water model during fixed protonation states and an implicit model for (de)protonation updates was deemed appropriate at the time.

If we were to conduct similar research in future, we may opt for an all-explicit water model in constant pH simulations, as was recently developed by Groenhoff and Hess groups and implemented in Gromacs (Aho et al., 2022: https://doi.org/10.1021/acs.jctc.2c00516). Motivated by this comment, we included a remark about the constant-pH simulation approach and the future direction in the 2.6 Limitations section of the paper.

p26, line 13: I much appreciate that the authors have made the effort to deposit the electronic material on Zenodo. This is a nice and very helpful gesture to the community!

We thank the reviewer for the kind acknowledgement!